# Assimilation of Pseudo-Tree-Ring-Width observations into an Atmospheric General Circulation Model

Walter Acevedo[1,2], Bijan Fallah[1], Sebastian Reich[2], and Ulrich Cubasch[1]

[1]Institut für Meteorologie, Freie Universität Berlin, Carl-Heinrich-Becker-Weg 6-10, 12165 Berlin, Germany.
[2]Institut für Mathematik, Universität Potsdam, Karl-Liebknecht-Strasse 24-25, 14476 Potsdam, Germany.

*Correspondence to:* Bijan Fallah (info@bijan-fallah.com)

**Abstract.** We investigate the assimilation of Tree-Ring-Width (TRW) chronologies into an atmospheric global climate model using Ensemble Kalman Filter (EnKF) techniques and a process-based tree-growth forward model as observation operator. Our results, within a perfect-model experiment setting, indicate that the nonlinear response of tree-growth to surface temperature and soil moisture does deteriorate the operation of the time-averaged EnKF methodology. Moreover, this skill loss appears significantly sensitive to the structure of growth rate function, used to represent the Principle of Limiting Factor (PLF)s within the forward model. On the other hand, it was observed that the error reduction achieved by assimilating a particular pseudo-TRW chronology is modulated by the strength of the yearly internal variability of the model at the chronology site. This result might help the dendrochronology community to optimize their sampling efforts. In our experiments, the "online" (with cycling) paleao Data Assimilation (DA) approach did not outperform the "offline" (no-cycling) one, despite its considerable additional implementation complexity.

## 1 Introduction

The low-frequency temporal variability of the climate system can not be estimated from the available time span of instrumental climate records. Accordingly, paleoclimate reconstruction must necessarily rely on the usage of the paleoclimate proxy records. Nonetheless, these natural archives exhibit several problematic features, e.g., low time-resolution, sparse and irregular spatial distribution, complex nonlinear response to climate and high noise levels. Therefore the proper extraction of the climate signal therein contained can often remain opaque (Evans et al., 2013).

At present, many different paleoclimate modeling ideas have been proposed, e.g., data-driven statistical techniques, climate model hindcasts, and Bayesian probabilistic methods (see Crucifix (2012) as a review). Among this plethora of approaches, DA methodologies are today particularly appealing as they allow to systematically combine the information of paleoclimate records with the dynamical consistence of climate simulations (Oke et al., 2002; Evensen, 2003; Hughes et al., 2010; Brönnimann, 2011; Bhend et al., 2012; Hakim et al., 2013; Steiger et al., 2014; Matsikaris et al., 2015; Hakim et al., 2016; Dee et al., 2016).

Heretofore, several very diverse paleo-DA schemes have been investigated providing very encouraging results (see (Hughes and Ammann, 2009; Widmann et al., 2010) as reference): Pattern Nudging (von Storch et al., 2000) and Forcing Singular Vectors (Barkmeijer et al., 2003; van der Schrier and Barkmeijer, 2005) techniques were designed to curb the atmospheric

circulation towards a target pattern by means of an artificial term added to the model dynamics. 4D-Var methodology has been used to assimilate pseudo-proxies into an ocean model (Paul and Schäfer-Neth, 2005; Kurahashi-Nakamura et al., 2014). EnKF was adapted to time-averaged observations (Dirren and Hakim, 2005) and tested for a hierarchy of atmospheric models (Huntley and Hakim, 2010; Bhend et al., 2012; Pendergrass et al., 2012; Steiger et al., 2014). Finally, the use of a particle filter

has been tested with an Earth system model of intermediate complexity (Annan and Hargreaves, 2012; Dubinkina et al., 2011; Dubinkina and Goosse, 2013; Mathiot et al., 2013).

A typical assumption in most of the paleo-DA studies so far conducted is that the climate-proxy relation is linear. Nonetheless, currently it is widely recognized that climate proxies are the result of complex recording processes, which can have physical, chemical and biological nature. Furthermore, several research groups have already developed and validated forward

models for several proxy types (Evans et al., 2013; Dee et al., 2015). Hence, in order to increase the realism of DA-based climate reconstructions, it is relevant and pertinent to connect the climate state space to the proxy space by way of forward models (Acevedo et al., 2015; Dee et al., 2016). Dee et al. (2016) applied three different nonlinear proxy system forward models in a DA framework and investigated the utility of paleoclimate observations for constraining climate simulations. They demonstrated that the linear-univariate models for tree ring width may not capture the GCM's climate, especially for regions

where the tree's growth is dominated by moisture. The tree ring forward model used in our study is a modified version of the model used in Dee et al. (2016). Acevedo et al. (2015) [AC15, hereafter] evaluated the applicability of the process-based TRW forward model Vaganov-Shashkin-Lite (VSL) (Tolwinski-Ward et al., 2011) as observation operator within a simplified DA setting. Using a chaotic 2-scale dynamical system as a toy model, AC15 generated pseudo-TRW observations and assimilated them via the time-averaged -EnKF algorithm (Dirren and Hakim, 2005). This paper follows closely the rationale of AC15, but

within a more realistic scenario, where an Atmospheric General Circulation Model (AGCM) is used as dynamical system and the observational network resembles the currently available TRW chronologies.

In addition to the classical DA approaches used in paleoclimate studies, a so-called "off-line" DA-based climate reconstructions is presented by (Steiger et al., 2014; Dee et al., 2016; Hakim et al., 2016). In an off-line approach the climate model is not re-initialized at the observation time steps (no initialization cycle or "no-cycling").

The main objectives of this study are to shed light on the following four fundamental questions :

1) Can paleo-DA improve the skill of the model for the forecast (prior) state?

2) Can paleo-DA improve the skill of the model for the analysis (posterior) state?

3) Can an on-line ("with cycling") DA outperform an "off-line" ("nocycling") one (see Sec.4 for the definition of "off-line")?

4) How does the nonlinear response of tree-growth to surface temperature and soil moisture affect the performance of the

30 time-averaged EnKF-DA method?

The third question is one of the most important challenges in the paleo-DA field, given that the computational expenses of an on-line DA scheme with a realistic coupled GCM is far beyond the affordable limits of today's computers. On the other hand, state-of-the-art climate models have little or no predictive skill on the long timescale of proxy records (Hakim et al., 2016).

In section 2 we describe the DA technique, the TRW forward model and the climate model as well as the experimental

setting used. Our numerical results are shown in section 3, followed by a discussion in section 4.

## 2 Materials and Methods

### 2.1 Data Assimilation Basics

#### 2.1.1 Rationale

The knowledge about the climate is drawn from observations and the physical laws governing the evolution of the climate
system. Numerical models apply the latter to estimate the state of the flow. DA is a process which applies both available
information sources to estimate the state of the climate (Talagrand, 1997).

In a typical *sequential* DA scheme, a climate model is integrated in time steps over which observations are available. The
predicted state at an observed instant (forecast), is used as "background" for the DA scheme. Furthermore, the forecast is
"updated" or "corrected" by observations to form the analysis. The model is then reinitialized from the analysis state and
propagates in time to reach the next observed instant. The analysis step is determined by availability of observations, their time
scales and computational resources. DA methods have evolved from very empirical approaches, such as Newtonian relaxation,
to probabilistic ones that estimate the state Probability Density Function (PDF) conditional to the observations (see Kalnay
(2003) and Lahoz et al. (2010) for review).

Among all the available DA techniques, EnKF (Burgers et al., 1998) offers an appealing trade-off between accuracy, rela-
tively user-friendly implementation and computational expenses. EnKF works robustly for very sparse observation networks
and moderate number of ensemble members (Whitaker et al., 2009). Its implementation does not require adjoint model (DA
calculations are outside the model code) and uncertainty estimates can be directly obtained from the ensemble spread (Hamill,
2006). The main disadvantage of EnKF, within a paleoclimate setting, is its inability to handle strongly non-Gaussian PDFs,
which can result from the nonlinearities of climate models and observation operators. Nonetheless, it is very difficult to re-
move this limitation, given that strictly non-Gaussian DA techniques have historically been prohibitively expensive to run for
high-dimensional systems.

#### 2.1.2 Kalman Filter

Given that the model's state is $\mathbf{x}(t) \in \mathbb{R}^n$, the Kalman Filter (KF) (Kalman, 1960) assumes that the PDF of forecast state is
given by a Gaussian function of mean $\mathbf{x}^f$ and covariance $\mathbf{P}^f \in \mathbb{R}^{n \times n}$:

$$p(\mathbf{x}) \propto \exp(-\frac{1}{2}(\mathbf{x} - \mathbf{x}^f)^T (\mathbf{P}^f)^{-1} (\mathbf{x} - \mathbf{x}^f)). \tag{1}$$

The observations $\mathbf{y}(t_j) \in \mathbb{R}^k$ are also assumed to have Gaussian errors and therefore the conditional probability of the
observation vector $\mathbf{y}$ given the state $\mathbf{x}$ is:

$$p(\mathbf{y} \mid \mathbf{x}) \propto \exp(-\frac{1}{2}(\mathbf{y} - \widehat{\mathrm{H}}\mathbf{x}^f)^T \mathbf{R}^{-1}(\mathbf{y} - \widehat{\mathrm{H}}\mathbf{x}^f)), \tag{2}$$

where $\widehat{H}$ and $\mathbf{R} \in \mathbb{R}^{k \times k}$ are the observation operator and the observation covariance matrix, respectively. Following the Bayes theorem, the conditional probability of the state given the observations, i.e., the analysis PDF, is:

$$p(\mathbf{x} \mid \mathbf{y}) \propto \exp(-\frac{1}{2}(\mathbf{x} - \mathbf{x}^f)^T (\mathbf{P}^f)^{-1}(\mathbf{x} - \mathbf{x}^f) - \frac{1}{2}(\mathbf{y} - \widehat{H}\mathbf{x}^f)^T \mathbf{R}^{-1}(\mathbf{y} - \widehat{H}\mathbf{x}^f)). \tag{3}$$

Assuming the $\widehat{H}$ is a linear function, equation 3 has also a Gaussian PDF (Eq.1 and Eq.2 are Gaussian). Therefore, its mean and covariance can be calculated by the so called Kalman update equations (Lorenc, 1986):

$$\mathbf{x}^a = \mathbf{x}^f + \mathbf{K}(\mathbf{y} - \widehat{H}\mathbf{x}^f), \tag{4}$$

$$\mathbf{P}^a = (\mathbf{I} - \mathbf{K}\widehat{H})\mathbf{P}^f; \tag{5}$$

where the Kalman gain matrix $\mathbf{K}$ is given by:

$$\mathbf{K} = \mathbf{P}^f \widehat{H}^\dagger (\widehat{H}\mathbf{P}^f \widehat{H}^\dagger + \mathbf{R})^{-1}. \tag{6}$$

### 2.1.3 Ensemble Kalman Filter (EnKF)

In a realistic model setting, the calculation of the covariance matrices are numerically very expensive. Evensen (1994) have used an ensemble of model states ($\mathbf{X}(t) = (\mathbf{x}_1, \ldots, \mathbf{x}_m)$) to approximate the KF equations. Following this approach the best state estimate and its uncertainty are presented by the ensemble mean and ensemble spread. The ensemble spread is given by the standard deviation of the ensemble around its mean. Thus, an EnKF cycle consists of an ensemble forecast step which provides the empirical mean and covariance for approximation of the KF equations:

$$\langle \mathbf{X}_f \rangle = \frac{1}{m} \sum_{i=1}^{m} \mathbf{x}_i^f, \quad \mathbf{P}^f = \frac{1}{m-1} \mathbf{X}_f{}'(\mathbf{X}_f{}')^T. \tag{7}$$

Here $\mathbf{X}_f{}' \in \mathbb{R}^{n \times m}$ denotes the forecast ensemble deviation matrix:

$$\mathbf{X}_f{}' = \mathbf{X}_f - \langle \mathbf{X}_f \rangle \mathbf{e}^T. \tag{8}$$

where $\mathbf{e} = (1, \ldots, 1) \in \mathbb{R}^m$. The analysis ensemble whose covariance satisfies equation 5 can be generated in different ways. Two main kinds of KFs are stochastic and deterministic filters (Hamill, 2006). In the stochastic approach an observational ensemble $\mathbf{Y}$ is generated by adding a set of realization of the observational noise to the observation vector $\mathbf{y}$. The analysis ensemble is created by the following updating equation:

$$\mathbf{X}_a = \mathbf{X}_f + \mathbf{K}\left(\mathbf{Y} - \widehat{H}\mathbf{X}_f\right). \tag{9}$$

In the deterministic updating scheme, instead of creating an ensemble of observations, the analysis mean ($\overline{\mathbf{X}_a}$) and deviations $\mathbf{X}_a{}'$ are calculated by using different update formula (Tippett et al., 2003).

Due to the limited ensemble size, the forecast uncertainty is usually underestimated in EnKF and after several assimilation cycles the observations may completely be ignored. This situation is known as "filter divergence" and can treated by multiplying the ensemble spread by a constant greater than one (covariance inflation).

For the experiments presented in this paper, we employed ensembles of 24 members (limited by the number of CPUs) and constant multiplicative inflation of $1\%$ after the ensemble update. As the consequence of limited ensemble size, any observation may present spurious correlations with the distant ones and the filter performance may be affected. Therefore, the elements of the observation error covariance matrix are multiplied by a function that increases exponentially with distance and an infinite error is assigned to the distant observations (R-localization (Hunt et al., 2007)). This is achieved using the following formula:

$$R_{loc} = R * \exp\left((r_h/2\lambda_h)^2 + (r_v/2\lambda_v)^2\right) \tag{10}$$

where $r_h$ and $r_v$ stand for the horizontal and vertical distances, respectively. Their corresponding scaling parameters were set to the values $\lambda_h = 500$ Km and $\lambda_v = 0.4 \ln p$.

### 2.1.4 Time-Averaged Ensemble Kalman Filter (EnKF)

Usually the time scale of the measured system is sufficiently longer than the response time of the sensor and the measurements can be assumed to be instantaneous. However, this assumption can not be applied for precipitation gauges, wind meters and proxy records. Proxies have averaged recording time spans ranging from months to decades. Time-averaged observations contain information of a segment of the model state trajectory instead of an instant of the model evolution.

The time-averaged ensemble background fields are updated by the EnKF and the instantaneous deviations from the mean remain unchanged. This approach is based on the fact that the observations can only contain time-averaged information (Dirren and Hakim, 2005).

### 2.1.5 Observational System Simulation Experiments

Given a prediction system comprising a dynamical model and a DA scheme, forecast and analysis errors arise from many different sources, e.g. model imperfections, inadequacy of the DA strategy and insufficiency of observational information, which interact with each other in practice. In order to disentangle the effects of these error sources, a DA scheme is typically tested under simplified conditions by means of numerical experiments, currently known as Observation System Simulation Experiments (OSSE), whose realism level is gradually increased.

An OSSE consists of (i) a single model trajectory $\mathbf{x}^{\text{NATURE}}$, typically referred to as "true" run or "nature" run, that is used as prediction target, (ii) pseudo-observations created by applying the observation operator to $\mathbf{x}^{\text{NATURE}}$ and adding simulated observational noise, and (iii) an observationally constrained run $\mathbf{X}^{\text{DA}}$, obtained by performing a sequence of analysis cycles where the pseudo-observations are assimilated (see Fig. 1).

The nature run is normally generated by running the dynamical model starting from a random sample of the model climatology. Notice that thanks to the availability of the truth model evolution for an OSSE, the forecast and analysis skill of the observationally constrained run can be directly assessed, using for example the Root Mean Square Error (RMSE) of the

ensemble mean:

$$\mathbf{RMSE}(\langle \mathbf{X}^{\text{DA}}\rangle) = \left( \overline{\left( \mathbf{x}^{\text{NATURE}} - \langle \mathbf{X}^{\text{DA}}\rangle \right)^2} \right)^{\frac{1}{2}}, \tag{11}$$

where $\overline{\phantom{x}}$ and $\langle\ \rangle$ denote the time and ensemble mean operators, respectively.

An additional run frequently performed for OSSE involving ensemble DA methods, is a free ensemble run $\mathbf{X}^{\text{FREE}}$, where no
observations are assimilated and then the ensemble just freely evolve under the action of the model dynamics. $\mathbf{X}^{\text{FREE}}$ is intended
to provide a benchmark of performance, against which it is possible to asses the the added value of the DA scheme.

## 2.2 TRW Forward Modeling

### 2.2.1 VSL Model

The VSL model for TRW chronologies offers an intermediate complexity approach between ecophysiological and completely
data-driven models (Tolwinski-Ward et al., 2011; Tolwinski-Ward, 2012), where the climate-driven component of tree-ring
growth is parametrized by way of a simple representation of the PLF (Fritts, 1976). This biological concept states that the pace
at which a plant develops is controlled by the single basic growth resource, typically either energy or water, that is in shortest
supply. Within VSL the limiting factors considered are near-surface air temperature ($T$) and soil moisture ($M$). These variables
influence tree growth by means of "growth response" functions $g_T$ and $g_M$ using a piece-wise linear "standard ramp" function
(Tolwinski-Ward et al., 2014):

$$\Psi(u) = \begin{cases} 0 & \text{if} \quad 0 \geqslant u \\ u & \text{if} \quad 0 < u \leqslant 1 \\ 1 & \text{if} \qquad u > 1, \end{cases}$$

VSL's growth responses at a particular time is expressed as:

$$g_T = \Psi\left( \frac{T - T^L}{T^U - T^L} \right) \tag{12}$$

and

$$g_M = \Psi\left( \frac{M - M^L}{M^U - M^L} \right). \tag{13}$$

Where $T^L$ and $M^L$ denote minimum thresholds for temperature and moisture below which there is no grown, and $T^U$ and
$M^U$ are upper thresholds above which tree growth is optimal. Afterwards, the growth rate $G_{MIN}$ is determined by the smallest
growth response, i.e.,

$$G_{MIN} = \min\{g_T, g_M\}, \tag{14}$$

The yearly TRW values $W$ are obtained as following:

$$W_n = \int\limits_{t_n-\tau}^{t_n} G_{MIN}(t)I(t)\,dt. \tag{15}$$

Where $I$ is the relative local insolation.

### 2.2.2 VSL from the Fuzzy Logic (FL) Viewpoint

The term FL was coined by Zadeh (1975) and refers to a mathematical theory which has been very successful at modeling complex systems involving imprecise data and vague knowledge of the underlying mechanisms. Since its introduction, FL has greatly influenced many applied disciplines, most notably control theory (Nguyen et al., 2002). Within the environmental sciences, FL has also found numerous applications, including ecological and hydrological modeling (Marchini, 2011; Salski, 2006; Se, 2009). Regarding climate proxy forward modeling, AC15 recently showed that VSL model can be completely

embedded into the framework of FL. Within this reinterpretation, the growth response function $g_T$ ($g_M$) correspond to the membership function to the set $S_T$ ($S_M$) of optimal temperature (moisture) conditions for tree growth. Temperature (moisture) values lying below $T^L$ ($M_L$) present null values for $g_T$ ($g_M$) and accordingly do not belong to $S_T$ ($S_M$). On the other hand, temperature (moisture) values lying above $T^U$ ($M_U$) lead to $g_T$ ($g_M$) values equal to 1, meaning they belong completely to $S_T$ ($S_M$). All the other temperature (moisture) conditions present growth responses between 0 and 1 and consequently they

are considered to belong partially to $S_T$ ($S_M$). This idea of partial membership is the basis of fuzzy logic and the sets defined this way are called fuzzy sets. Furthermore, the intersection of the fuzzy sets $S_T$ and $S_M$ is again a fuzzy set $S_{T\wedge M}$, whose membership function can be calculated by evaluating the minimum between $G_T$ and $G_M$:

$$g_{T\wedge M} = \min\{g_T, g_M\} \tag{16}$$

Equation 16 is completely equivalent to the equation 15 and then VSL's growth rate function can be interpreted as the

membership function for the fuzzy intersection set $S_{T\wedge M}$. In FL theory, the minimum function (Eq. 16) is one of the most popular representations of the intersection operation, however it is not the only, existing actually a whole family of appropriate functions referred to as t-norms (see Nguyen et al. (2002)). In AC15 a number of t-norms was tested as replacement for VSL's growth rate function within a highly simplified paleo-DA setting. In particular it was found that the product t-norm $g_{T\wedge M} = g_T \cdot g_M$ might improve significantly the performance of the time-averaged EnKF technique. Accordingly, beside the

minimum t-norm we consider also in this paper the product growth response VSL with Product t-norm (VSL-Prod):

$$G_{PROD} = g_T \cdot g_M. \tag{17}$$

## 2.3 Atmospheric General Circulation Model

The Simplified Parametrizations, primitivE-Equation Dynamics (SPEEDY) model (Molteni, 2003) is an intermediate complexity AGCM comprising a spectral dynamical core and a set of simplified physical parametrizations, based on the same principles as state-of-the-art AGCM but tailored to work with just a few vertical levels.

SPEEDY's dynamical core solves the hydrostatic primitive equations by means of the spectral transform developed by Bourke (1974), which uses absolute temperature, logarithm of the surface pressure, specific humidity, divergence and vorticity as basic prognostic variables. The time stepping is performed via a leapfrog scheme with an standard Robert–Asselin filter (Robert, 1966). The sub-grid scale processes parametrized in speedy are convection, large-scale condensation, clouds, short- and long-wave radiation, surface fluxes, and vertical diffusion.

In this paper we employ version 32 of SPEEDY, with seven vertical levels (L7) and standard Gaussian grid of 96 by 48 points in the horizontal (T30). The top and bottom layers represent the stratosphere and the planetary boundary layer, respectively. Despite of its low resolution and the relative low complexity of its parametrizations, SPEEDY still captures many observed global climate features in a realistic way, while its computational cost is at least one order of magnitude lower than the one of sophisticated state-of-the-art AGCM's at the same horizontal resolution (Molteni, 2003). The latter makes SPEEDY specially suitable for studies involving long ensemble runs, like the ones necessary for this study.

## 2.4 Experimental Setting

Following the rationale used in the experiments of AC15, pseudo-TRW observations are generated using Vaganov-Shashkin-Lite (VSL) (Tolwinski-Ward et al., 2011, 2013) as observation operator. Afterward, the time-averaged state of the atmosphere is estimated via EnKF approach Dirren and Hakim (2005). The impact of the representation of the PLF on the filter performance is studied using the assimilation of time averaged linear observations as a reference.

### 2.4.1 Filter Implementation

The SPEEDY model is embedded by Miyoshi (2005) into the ensemble DA framework using the Local Ensemble Transform Kalman Filter (LETKF) (Hunt et al., 2007), the so called SPEEDY-LETKF framework. The parallel FORTRAN 90 implementation of the LETKF is promising for high resolution model given that the calculation of the analysis for a particular grid point requires only the information of the neighboring grid points. Therefore, LETKF offers outstanding scalability properties. SPEEDY-LETKF is an open-source software which have already been widely used for several DA studies (Li et al., 2009; Miyoshi, 2010; Lien et al., 2013; Ruiz et al., 2013; Amezcua et al., 2014). Here, SPEEDY-LETKF was extended for the assimilation of time averaged linear observations and pseudo-TRW observations. This was done by (i) modification of the model time cycling, (ii) addition of the time-averaged updating approach of Dirren and Hakim (2005) and (iii) development of the VSL-like observation operator.

Additionally, in order to avoid catastrophic filter divergence (ref. sec. 2.1.3), observations with large divergence from their corresponding predicted values were neglected. Moreover, in order to avoid the crash of the model after assimilation steps,

the following quality control criterium is applied: The observations whose corresponding innovation vector norm (absolute mismatch regarding the forecast observation) is bigger that 10 times its error standard deviation are discarded.

### 2.4.2 Simulations' Characteristics

The modified version of SPEEDY-LETKF is utilized to carry out a set of standard "perfect model" OSSEs (Fig. 1). First the VSL with Temperature t-norm (VSL-T) representation of the PLF is utilized for two sets of experiments under different ocean conditions:

• *PRESCRIBED experiment* is forced by the boundary conditions included in the version 41 of the code, which comprises the sea surface temperature (SST) anomalies from 1854 to 2010 with respect to the period 1979 to 2008 derived from NOAA_ERSST_V3 dataset (Smith et al., 2008; Xue et al., 2003), as well as climatological maps derived from input data of the European Centre for Medium-Range Weather Forecasts (ECMWF)'s reanalysis (Gibson et al., 1997). At the surface boundaries model requires the climatological maps of sea surface temperature, sea ice fraction, surface temperature at the top of the soil, moisture in the top soil layer and the root-zone layer, snow depth, bare-surface albedo, fraction of land-surface vegetation. At the top of the atmosphere, the model calculates the flux of incoming solar radiation from astronomical formulae (Molteni, 2003). The solar radiation absorption by ozone in the stratosphere follows empirical functions with seasonal variability. The latitudinal variability of the optical depth depends on the daily averaged zenith angle (Molteni, 2003). The climatological fields are derived for the period 1981-1990 to have a better balance for warm and cold El Niño-Southern Oscillation (ENSO) events (Molteni, 2003). This procedure follows the AMIP-type experiments (Herceg Bulić and Kucharski, 2012).

• *SLAB experiment* is coupled with a slab ocean model ("q-flux adjusted mixed layer model") forced by climatological ocean dynamics and no initialization is used. The model starts from a spun up state.

Two representations of the PLF are considered: the "minimum" ($G_{MIN}$) and the "product" ($G_{PROD}$) t-norms. Initially, a one-year long spin-up run is performed for all experiments, starting from January $1^{st}$, 1860. The final state of this model trajectory is subsequently used as initial condition for a 150 year long nature ("true") run. The ensemble runs with and without DA are identically initialized from a set of states gathered daily from the last two months of the spin-up run (lagged 2 day initialization). Notice that the nature ("true") run and the different ensemble runs (priors) are generated with the same time varying forcing fields.

### 2.4.3 Observation Generation

Pseudo-TRW observations are produced following VSL's formulation, plus a final white noise addition step, where random draws from a Gaussian distribution are imposed on the time averaged observations. The measurements' error is assumed not to be correlated in time (no memory), therefore the white noise is used in this study (McShane and Wyner, 2011; Dee et al., 2016). Surface temperature data was extracted from the lowest level of the state vector, while soil moisture was taken from the surface boundary conditions. Notice that temperature is a prognostic variable of the model, whereas soil moisture is a prescribed variable with yearly periodicity. It is worthwhile to mention that although soil moisture is not a prognostic variable of SPEEDY, it does affect prognostic variables, such us humidity, through the parametrizations.

Regarding the geographical distribution of observations, we place a station at every grid box where at least one actual TRW chronology from the database of Breitenmoser et al. (2014) is present. This strategy yields an observational network comprising 257 stations (see figure 2). Concerning the configuration of the observation operator, for our experiment involving SPEEDY we focus on the effect of the first VSL's nonlinearity, i.e., the shifting of recorded variable. Consequently, we configure VSL so that no thresholding takes place. This is done by setting the upper and lower response thresholds to the maximum and minimum values during the nature (true) run, respectively, so that the response functions reduce to linear rescaling operators (ref. AC15).

### 2.4.4 Diagnostic Statistics

Our results are presented in three sections: 1) time-series of globally averaged temperature RMSE, 2) histograms of these time-series and 3) maps of time-averaged (150 years) temperature RMSEs. We show the analysis of the temperature variable due to its larger error reduction compared to other variables (*eg,* humidity, u-wind, v-wind) when DA is applied.

## 3   Results

Given the annual resolution of TRW chronologies, we study the filter performance for yearly averaged values (near surface temperatures). We monitor the behavior of ensemble runs by means of RMSE for the near surface temperature. SPEEDY presents spatially heterogeneous internal variability (Molteni, 2003). Due to this feature, for a particular time averaging length, there will typically be regions with very low internal variability (*eg.,* equatorial regions for temperature) for which RMSE shows very low values.

### 3.1   Free Ensemble Run

An AGCM is an example of non-autonomous system and accordingly the evolution of its state is determined by both the atmospheric dynamics and the external forcing. The influences of these two distinct factors can be disentangled to some extent by considering atmospheric variability to be a superposition of an internal component, caused by the intrinsic dynamics, and an external one, resulting from the variations of the boundary conditions (Deza et al., 2014). Under this assumption, internal and external variability can be separated by way of a free ensemble run, using the ensemble mean as an estimate of the forced component. The magnitude of the internal variability can then be estimated from the ensemble spread. Note that using an ensemble DA method is only beneficial in the presence of internal variability, given that the forced variability can be well described by an unconstrained ensemble run (free ensemble run).

### 3.1.1   Free Ensemble Spread and Error

The time averaging operator acts as a low pass filter that reduces the amplitude of fluctuations with time scales shorter than the averaging period. Subsequently, geographical areas dominated by fast processes, compared to averaging period, tend to present constant mean values, or equivalently no internal time averaged variability. The climatology and the formulation of the SPEEDY model is fully described in Molteni (2003). Therefore, we focus only on the results of the DA approach, without

considering the systematic errors of the model. In the case of TRW chronologies, the characteristic one-year averaging period is long for atmospheric phenomena, and as consequence several areas show very low yearly internal variability for certain variables. A clear example of this is temperature around the equator (see figure 3a) where the temperature variability is dominated by the daily cycle and accordingly is strongly attenuated by the yearly averaging. On the other hand, planetary scale patterns are
not completely stationary and fluctuate over longer time scales. These low-frequency processes introduce internal variability in the yearly means, as can be seen in figure 3a. Maximum temperature spread occurs near the surface at high latitudes around $\pm 70°$. These yearly internal variability maxima can be related to leading modes of variability of the global circulation, such as the "annular modes" (e.g., ENSO) (Thompson and Wallace, 2000), migrations of the Inter Tropical Convergence Zone (ITCZ) (Schneider et al., 2014), as well as displacements of the jet stream (Woollings et al., 2011).
An important consequence of the spatially heterogeneous yearly internal variability of SPEEDY is that the nature (true) run variables at geographical areas with low internal variability can be well predicted by the ensemble mean of the free ensemble run, as it can be seen in figure 3b for the tropical surface temperature. On the other hand, RMSE extremes take place in areas of maximal internal variability (compare figures 3a and 3b). Generally, the error of the free ensemble run, used as a predictor of the nature (true) run, is essentially the projection of the nature (true) run trajectory on the internal variability component
(see schematic in Fig. 1). Figure 3 exhibits the results for the SLAB experiment. The PRESCRIBED experiment presents very similar behavior.

### 3.2  Assimilating Pseudo-TRW Observations

### 3.2.1  Temperature-based PLF Representation (VSL-T)

As described in Sec. 2.4.2, we investigate two different experiments using SLAB and PRESCRIBED ocean conditions (see
Table 1). For the sake of simplicity, we set up the sensitivity experiments using the simple observation operator VSL-T to investigate the effect of the SLAB ocean model. The use of SLAB ocean is motivated by the fact that the coupled ocean may lend predictability to the atmosphere as a slow component of the climate system. On the other hand, the climate of the PRESCRIBED experiment may follow the trends of the forced ocean conditions instead of the terrestrial proxy records. Therefore, the PRESCRIBED experiment's spread and error are expected to be smaller than the SLAB experiment. Figure 4
supports this hypothesis, showing a reduction in globally averaged free ensemble error in PRESCRIBED compared with the SLAB.

Figure 4a illustrates that no error reduction is obtained for the forecasted temperatures. The expected value of the RMSE is slightly larger than the free ensemble simulation for both SLAB and PRESCRIBED. However, the analysis state has skill (Fig. 4b), especially prior to 1950s. The existence of the trend in the RMSE time-series may arise from cycling (reinitialization)
of the ensemble or the choice of observation operator (more details in Sec. 4.2). The distribution density of the proxy record locations are biased to the Northern hemisphere; therefore, the error reduction of the analysis is more evident in the RMSE maps (Fig. 5).

An important aspect of our results concerning the DA skill when yearly averaged linear observations are assimilated, is that the error reduction regarding the free ensemble run appears modulated by the magnitude of the yearly internal variability of the particular variable at a specific site (compare figures 4 and 5). As a consequence, stations located in areas of strong yearly internal variability are more efficient than the others at reducing the error of the time-averaged state estimate. An example of this are the stations located in Alaska which constrain temperature with considerably larger skill than TRW sites in South Africa. This finding may prove useful for the design of optimal TRW chronology networks, in particular, and proxy networks in general (see Comboul et al. (2015) and the discussion at the end of this paper).

This situation, where a DA method presents time averaged analysis skill for averaging periods where the time averaged forecast skill is completely lost, has been previously observed in studies applying EnKF techniques on time averaged quantities (Huntley and Hakim, 2010; Bhend et al., 2012; Pendergrass et al., 2012; Steiger et al., 2014). DA performed under these circumstances is generally termed "offline Data Assimilation". This term is used to indicate that, under the randomizing action of chaotic model dynamics, the prior is completely decorrelated from the previous analysis state. As a consequence, the observational information cannot accumulate over time, as opposed to the typical application of DA for short-range prediction. This complete absence of observational constraint on the forecast implies that our DA experiments are performed in an "off-line" regime.

### 3.2.2 Minimum and Product Growth Rate Functions (VSL with Minimum t-norm (VSL-Min) and VSL-Prod)

Here the performance of the two different growth functions within the VSL's formulation, the product growth response ($G_{Prod}$) and the minimum growth response ($G_{Min}$), are investigated. These formulations are tested for both "online" (with cycling) and "offline" (no-cycling) data assimilation set-ups. In a simple DA experiment, AC15 have shown that the $G_{Prod}$ performs slightly better than $G_{Min}$.

#### 3.2.2.1 DA with cycling

Considering the SLAB experiment, we compare the two nonlinear PLF representations in our DA setting (VSL-Min and VSL-Prod). As illustrated in figure 6a, the DA forecast presents no skill in the globally averaged temperature for both of the representations. However, the use of VSL-Prod, instead of VSL-Min appears beneficial to the filter performance for the analysis, as demonstrated in Figure 6b. The expected value of the RMSE shifts significantly toward lower values for VSL-Prod compared to the free ensemble run. Similar to the case of VSL-T, the RMSE time-series shows an increasing trend for both VSL-Min and VSL-Prod.

The RMSEs of DA forecasts using different VSL representations (figures 5a, 7a and 8a) do not indicate any improvement over the free ensemble run (Fig. 3b). The analysis of VSL-Prod performs with slightly better skill than VSL-Min over Europe, the United States and Central Asia. Due to the strong nonlinear features of VSL-Min and VSL-Prod, the performance of these filters is expected to be degraded with respect to the ensemble runs constrained with VSL-T linear observation (see AC15). This behavior can be readily seen by comparing the figures 5b, 7b and 8b.

#### 3.2.2.2 DA with no-cycling

Our experiments show that the DA forecast has no skill over the model climatology. Several recent studies have applied a similar DA methodology (Steiger et al., 2014; Dee et al., 2016; Hakim et al., 2016). Dee et al. (2016) have performed paleoclimate reconstructions by using a physically based Proxy System Models (PSM) for three kinds of proxies (tree ring, coral $\delta^{18}O$ and ice core $\delta^{18}O$) and two isotope-enabled atmospheric general circulation models. Matsikaris et al. (2015) compared an off-line and an on-line ensemble-based DA and showed that the both methods outperform the model without DA. They concluded that the on-line method performs a more realistic temporal variability. Therefore, we investigate the idea of purely "off-line" DA. The free ensemble simulation at any individual year is used as the prior state vector for that year instead of the DA forecast. Following this methodology, the cycling step (reinitialization) of the ensemble is neglected and the time averaged DA technique is applied in parallel. A very interesting feature of figure 9 is that the increasing trends in the RMSE time-series of the analysis have vanished. This indicates that the previously existing trends in the forecast and consequently in the analysis originated from the reinitialization step of the system but not the proxy records. Figure 10 also confirms that the performance of no-cycling DA can compete with the performance of the online DA.

### 3.2.3 Signal to Noise Ratio

The Signal to Noise Ratio (SNR) is expressed as the ratio of the standard deviation of the nature (true) run to that of the additive white noise. We examined the performance of the off-line DA with different SNRs (Fig. 11). Figure 11 exhibits that the time-averaged global RMSE shows an elbow at values around $SNR = 1$ and reaches the error levels of Free run at $SNR = 0.03$, where all the pseudo-observations are ignored in the DA.

### 3.2.4 Time Variable Soil Moisture

To investigate the effect of using the time-varying soil moisture fields instead of climatological average in DA approach, we implemented the Climate Prediction Center (CPC) Leaky Bucket Model (LBM) (Huang et al., 1996) in our DA code. The LBM code was extracted from VSL v2_3 (ftp://ftp.ncdc.noaa.gov/pub/data/paleo/softlib/vs-lite/). Instead of using climatological soil moisture for VSL, the precipitation and temperature output from SPEEDY is used as input for LBM to produce the new set of soil moisture with interannual variations. In the next step we repeated the off-line data assimilation runs for two VSL presentations (VSL-Prod and VSL-Min).

The results show that using the new set of soil moisture has improved the error reduction of VSL-Min with minor improvement for VSL-Prod in both time evolution and maps of RMSE (Figures 12 and 13). The RMSE of VSL-Min reaches the one of VSL-Prod when using the soil moisture calculated from LBM. Figure 14 shows the histograms of the RMSE time-series. The results show that the VSL-Min is more sensitive to the choice of soil moisture and using the soil moisture calculated by the LBM improves the performance of the model. However, the improvement in error reduction for VSL-Prod is not significant when using the calculated soil moisture with the LBM.

## 4 Discussion and Conclusions

### 4.1 Error Reduction Efficacy of TRW Chronologies

For the OSSEs studied here, it was found that the ability of a particular pseudo-TRW chronology to reduce the error of the EnKF-based estimate of the time averaged state appears modulated by the strength of the yearly internal variability of the model at the chronology site. This methodology is termed Optimal Sensor Placement (OSP) and can in principle be employed to help the dendrochronology community to increase the effectivity of their sampling efforts by focusing on the sites with more potential to decrease reconstruction uncertainty (Ancell and Hakim, 2007; Hakim and Torn, 2008; Mauger et al., 2013). Furthermore, this approach can be directly applied to any proxy type with sufficiently stable time resolution (e.g., annual resolution)(Comboul et al., 2015). However, the application of this method for lower frequency climate data like sediment cores or speleothems has to be investigated. These results are likely to depend on the climate model, the proxy system model, the proxy network and their resolution (Comboul et al., 2015).

An evident caveat of the above mentioned rationale is that every model-based estimate of the climate internal variability strength for a particular time scale will necessarily exhibit the biases of the particular climate model used. We consider that this modeling subjectivity/imperfection issue can be ameliorated by means of multi-model and multi-physics approaches, which in principle should increase the robustness of the results and provide uncertainty estimates. In any case, we believe that provided results are analyzed cautiously taking into account the weaknesses of current climate models. The climate dynamics knowledge condensed into an Earth system model can certainly be used profitably to reduce the cost of a indiscriminated proxy sampling strategy.

### 4.2 Off-line Regime

Within our simplified perfect model OSSE, the observed situation of simultaneously having significant DA skill for analysis quantities and none for forecast quantities, currently referred to as off-line DA regime, can arise either from the dynamical model or from the DA scheme (answers to the questions 1 and 2 raised in Introduction).

Regarding the dynamical model, the most obvious reason to enter into the off-line regime is that the period between consecutive observations exceeds the predictability horizon of the model. Under these conditions, as already discussed in AC15, the ensemble spread reaches climatological levels (spread of the Free ensemble run) before new observations are assimilated and the accumulation of observational information is essentially lost. For SPEEDY, due to its purely atmospheric nature, it is likely to enter the off-line regime for a 1-year inter-observation period. This might be also the case for current operational (coupled) climate prediction systems, given their lack of useful lead times longer than one year. Thus, it seems unlikely to achieve effective observational constraints on the forecast using proxy records with yearly time resolution. However, there is already evidence for the existence of potential sources of climate internal variability with time scales longer than 1 year (Smith et al., 2012). The so called "annular modes" (Thompson and Wallace, 2000) may present internal variability in the high latitude areas. The latitudinal oscillation of the cell structure imposes variability at the fringes of the jet streams and oscillations of the

ITCZ impacts the humidity (Holton and Hakim, 2013). ENSO affects a large portion of tropical and subtropical climate in time-scales larger than one year. Accordingly, we expect that it should be possible to obtain actual inter-annual predictability skill in the foreseeable future.

Regarding the DA scheme, a possible culprit for the onset of the off-line regime is the time-averaged update strategy (Dirren and Hakim, 2005). It is not clear if whether we can employ this technique with SPEEDY to properly estimate instantaneous quantities out of time averaged observations. In particular, complete decorrelation between time averaged and instantaneous variables is not guaranteed.

In any case, despite its lack of accumulation of observational information over time, off-line DA has already been shown to be more robust than traditional Climate Field Reconstruction (CFR) techniques based on orthogonal empirical functions and stationarity assumptions (Steiger et al., 2014; Hakim et al., 2016). Moreover, the implementation and running of off-line DA schemes is remarkably cheaper than on-line approaches.

Following the idea of (Steiger et al., 2014; Matsikaris et al., 2015; Hakim et al., 2016) for purely "offline" DA (no-cycling), our perfect model experiments indicate that the "online" scheme may not outperform the "offline" one in either the temporal or the spatial error reduction (answer to the question 3 raised in Introduction). It should also be emphasized that our model set-up (with slab ocean) can not capture the full atmosphere-ocean interactions. Therefore, using a more realistic coupled atmosphere-ocean model may improve the skill of the "online" DA.

### 4.3 Filter Operation Sensitivity to the Growth Rate Function

The results of the DA experiments conducted with SPEEDY support results obtained the two-scale Lorenz (1996) model (AC15) regarding the influence of the PLF representation on the filter performance. The efficacy of the EnKF-based time averaged state estimation strategy appeared to be significantly sensitive to the selection of the t-norm used to calculate the growth rate, with the product t-norm (VSL-Prod) outperforming the minimum t-norm (VSL-Min) used in the original formulation of VSL forward model.

Tolwinski-Ward et al. (2014) described trees as fundamentally lossy[1] recorders of climate, due to the integrated nature of the information contained in them and the standardization process used to minimize the non-climatic effects on growth. Growth is influenced by temperature and/or moisture and the transitions between limitation regimes may happen suddenly ("abrupt shifting") Acevedo et al. (2015). In the same vein, we argue that the "abrupt shifting" of recorded variable (temperature or moisture)–implied by the minimum function used in VSL's original formulation– might constitute an additional source of lossyness (at least within a EnKF-based DA setting used), which can be substantially reduced by resorting to alternative Fuzzy Logic-based representations of the PLF. Our DA experiment indicates a higher skill performance with the VSL-Prod for both "offline" and "online" regimes compared to the VSL-Min (answer to the question 4 raised in Introduction).

---

[1]This adjective is currently used in the information technology area to designate data encoding methods that lead to information loss from the original version for the sake of reducing the amount of data needed to store the content.

## 4.4 Challenges to be Addressed

As a cautionary remark, we want to highlight the several important limitations of the experiments described in this paper. The generated pseudo-TRW observations lack a threshold for temperature or moisture after which the growth response does not change and their contamination with noise was performed assuming optimistically high SNR levels. Furthermore, the response thresholds were set in a completely homogeneous fashion for all the observational stations, whereas actual TRW networks are strongly heterogeneous in that sense, comprising chronologies generated under highly dissimilar growth limitation regimes. Additionally, the efficiency of EnKF technique used relies on the Gaussianity of all the variables of the model. Nevertheless, in a climate model some variables can present strongly non-Gaussian properties –specially definite positive quantities such as humidity– and their estimation should in principle be performed with more sophisticated strategies such a Gaussian anamorphosis (Bocquet et al., 2010; Lien et al., 2013). It is worth mentioning the necessity of explicitly addressing model errors by conducting imperfect model OSSE. Finally, we note that our findings are based on a slab coupled ocean model and we encourage using a proper coupled atmosphere-ocean model in future studies.

## 4.5 Outlook

Our results appear useful for TRW chronologies in the sense that EnKF techniques are robust in the face of two strong non-linearities, i.e., "switching recording" (Acevedo et al., 2015). Thus, it is important to emphasize that the OSSE presented in this manuscript represents the first step of the long hierarchy of DA experiments to achieve an effective assimilation of proxy records into climate models using forward proxy models. We encourage further experiments using comprehensive earth system models with longer time scale processes to bring the proxy DA into an online regime. However, assimilation of proxies in an earth system model with different components may lead to inter-component DA pollutions.

*Acknowledgements.* This work was supported by German Federal Ministry of Education and Research (BMBF) as Research for Sustainability initiative (FONA); www.fona.de through PalMod project (FKZ: 01LP1511A). The computational resources were made available by the High Performance Computing Center (ZEDAT) at Freie Universität Berlin and the German Climate Computing Center (DKRZ). W.A. wishes to acknowledge partial financial support by the Helmholtz graduate research school GEOSIM.

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

**Table 1.** Runs' characteristics.

| *No.* | 1 | 2 | 3 | 4 |
|---|---|---|---|---|
| **Forward Model** | VSL-T | VSL-T | VSL-Min | VSL-Prod |
| **Ocean** | *SLAB* | *PRESCRIBED* | *SLAB* | *SLAB* |

Simulations are 150 years long.

Whitaker, J. S., Compo, G. P., and Thépaut, J.-N.: A Comparison of Variational and Ensemble-Based Data Assimilation Systems for Reanalysis of Sparse Observations, Mon. Wea. Rev., 137, 1991–1999, http://dx.doi.org/10.1175/2008MWR2781.1, 2009.

Widmann, M., Goosse, H., van der Schrier, G., Schnur, R., and Barkmeijer, J.: Using data assimilation to study extratropical Northern Hemisphere climate over the last millennium, Climate of the Past, 6, 627–644, doi:10.5194/cp-6-627-2010, http://www.clim-past.net/6/627/2010/, 2010.

Woollings, T., Pinto, J. G., and Santos, J. A.: Dynamical Evolution of North Atlantic Ridges and Poleward Jet Stream Displacements, J. Atmos. Sci., 68, 954–963, http://dx.doi.org/10.1175/2011JAS3661.1, 2011.

Xue, Y., Smith, T. M., and Reynolds, R. W.: Interdecadal Changes of 30-Yr SST Normals during 1871-2000, J. Climate, 16, 1601–1612, doi:10.1175/1520-0442(2003)016<1601:ICOYSN>2.0.CO;2, http://journals.ametsoc.org/doi/abs/10.1175/1520-0442%282003%29016%3C1601%3AICOYSN%3E2.0.CO%3B2, 2003.

Zadeh, L. A.: Fuzzy logic and approximate reasoning, Synthese, 30, 407–428, 1975.

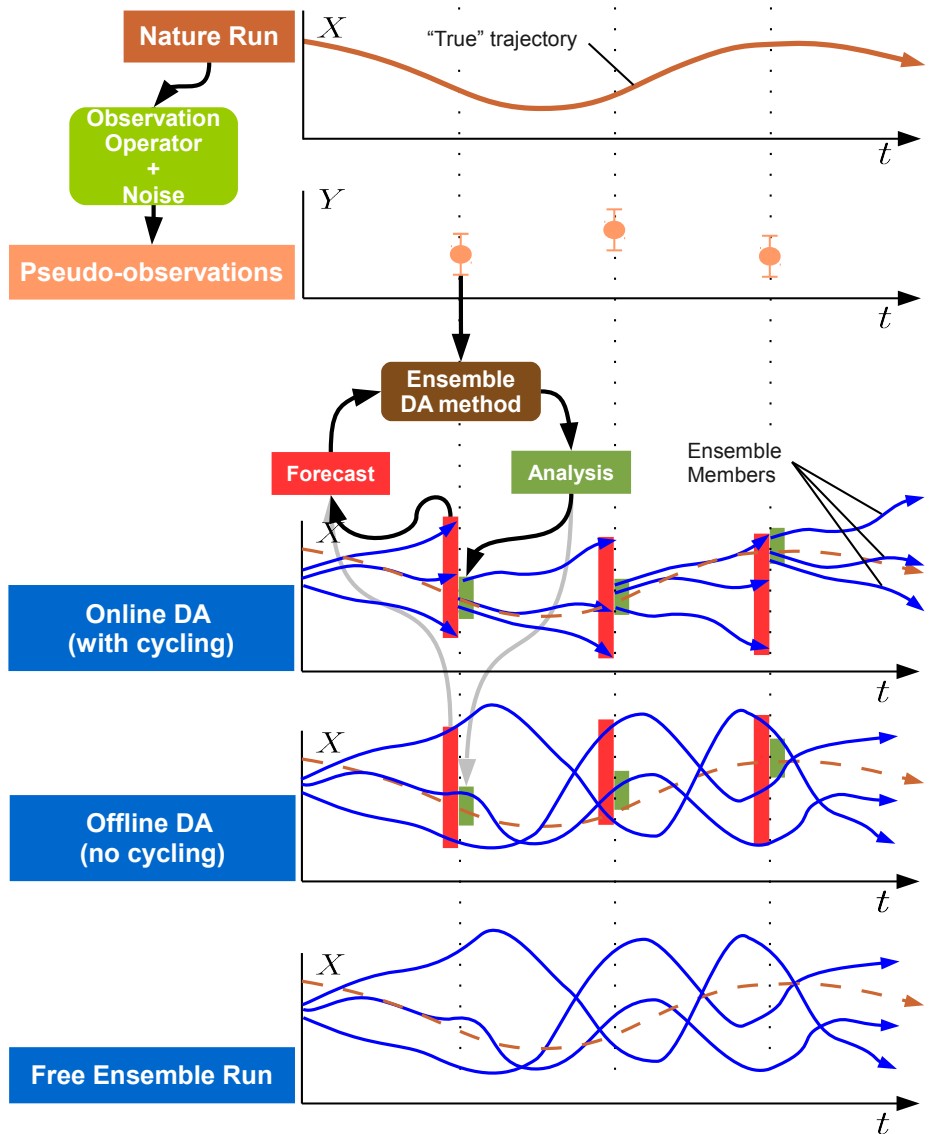

**Figure 1.** Schematic of a typical Observation System Simulation Experiment (OSSE) with ensemble "online" (with cycling) and "offline" (no-cycling) DA methods. $t$ designates the time axis and $X(Y)$ denotes the model state (observation) space. Sharp (rounded) cornered boxes represent data (processes). Red (green) vertical shadings indicate the *Forecast* (*Analysis*) spread. Vertical dotted lines represent the assimilation steps.

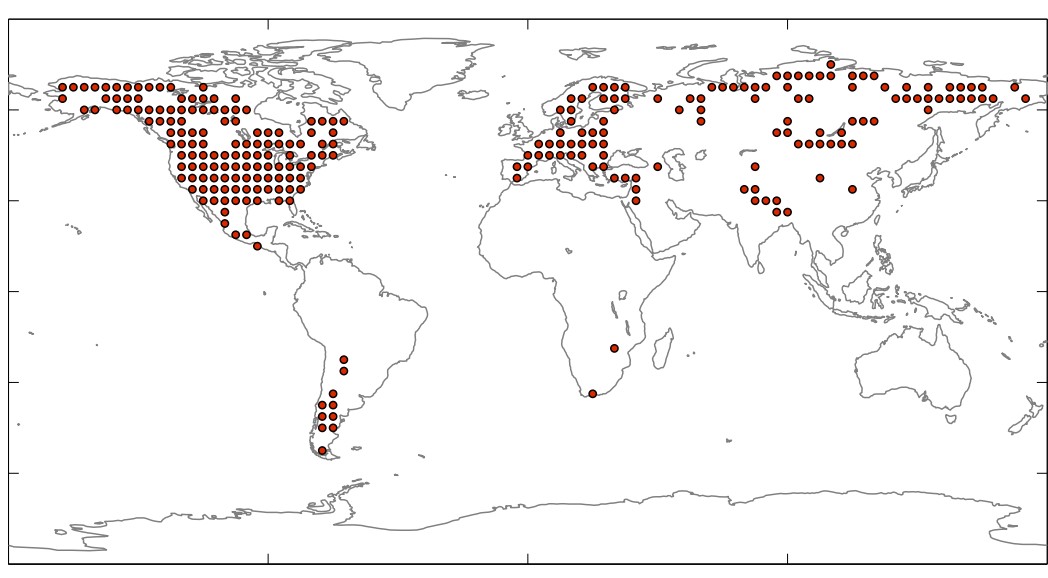

**Figure 2.** Station set resembling real TRW network from Breitenmoser et al. (2014)

**a) Free ensemble Spread**

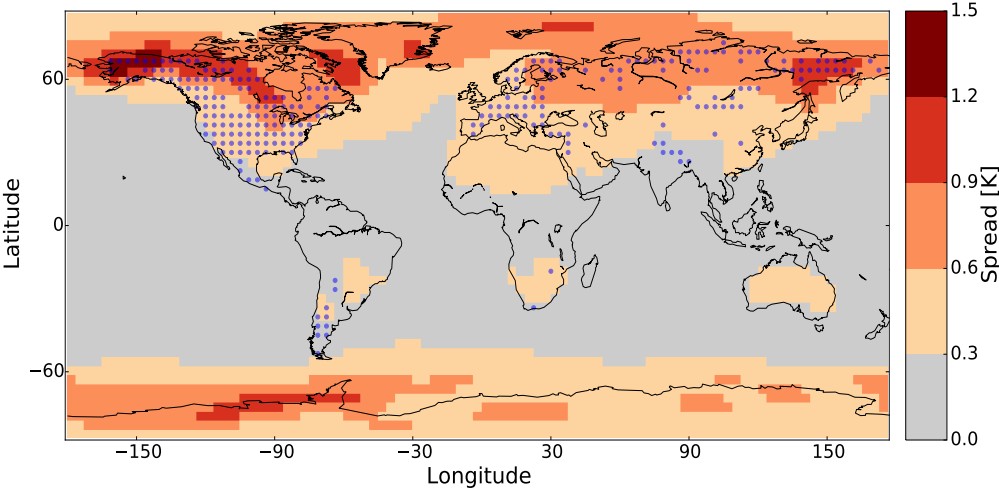

**b) Free ensemble Error**

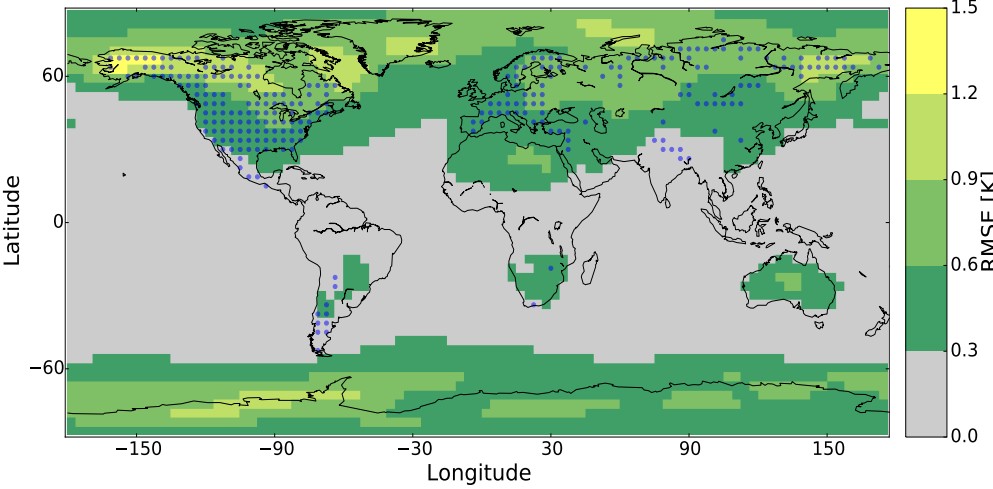

**Figure 3.** Free ensemble simulations for the SLAB experiment: a) Ensemble Spread [K] of near surface temperatures, b) Free ensemble RMSE [K].

a)

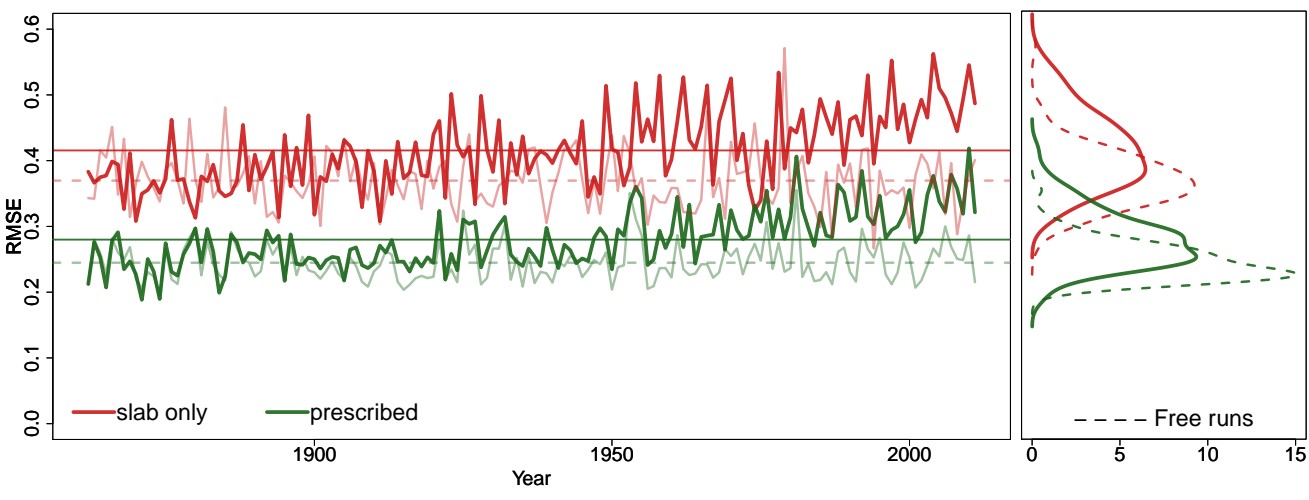

b)

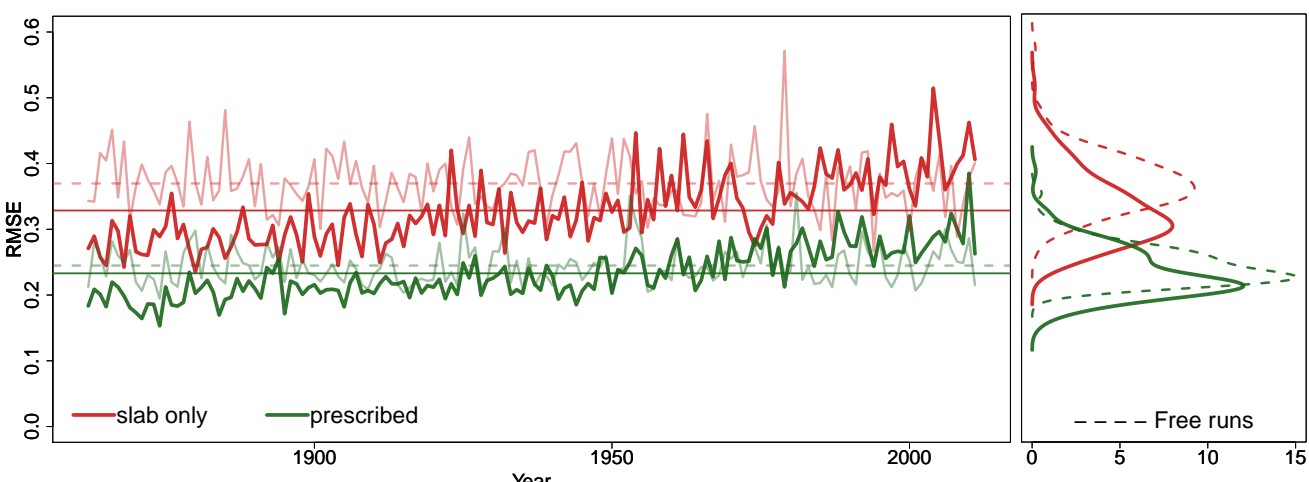

**Figure 4.** Global ensemble mean for a) Forecast constrained by VSL-T pseudo-TRW observations (bold lines) and Free run (thin lines); b) Analysis (solid lines) and Free run (thin lines). Horizontal lines exhibit the mean values. Right panels exhibit the histograms of the time-series.

**a) DA forecast for VSL-T**

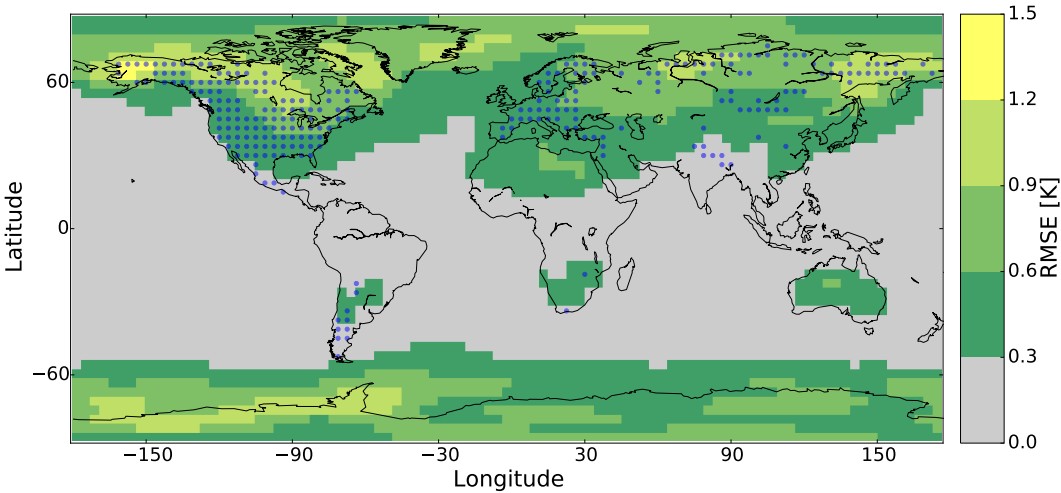

**b) DA analysis for VSL-T**

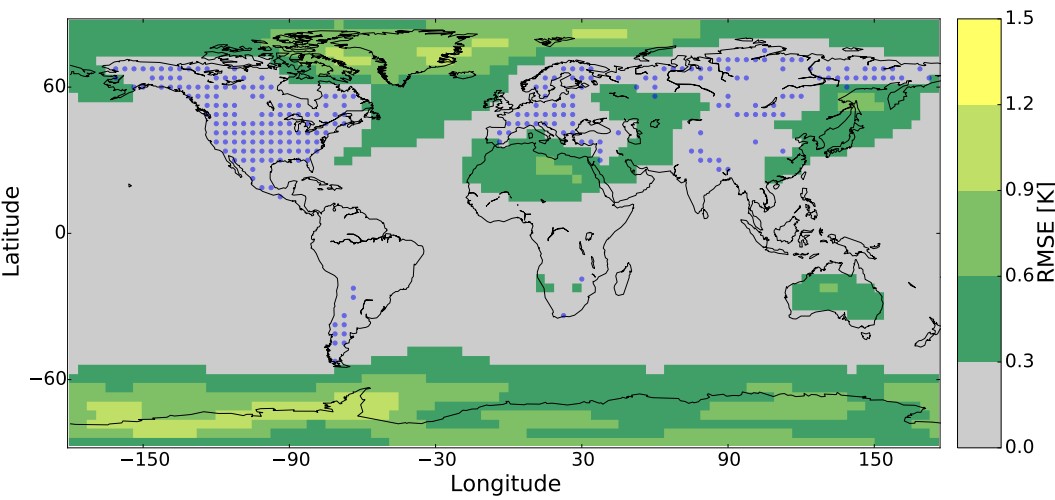

**Figure 5.** Time-averaged RMSEs of SLAB experiment for a) DA forecast and b) DA analysis using the VSL-T observation operator.

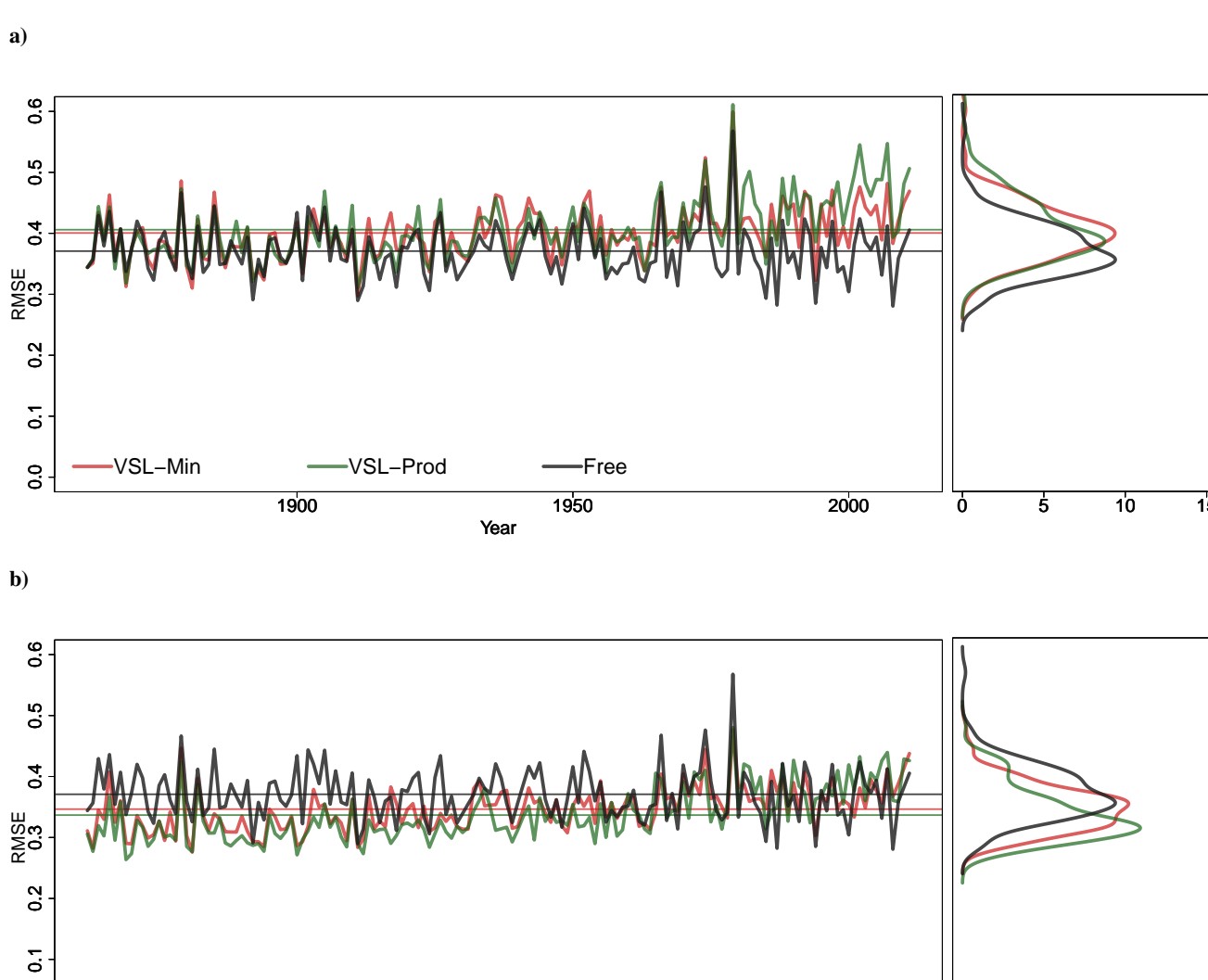

**Figure 6.** Global ensemble mean for a) forecast and b) analysis constrained by VSL-Min (red) and VSL-Prod (green) pseudo-TRW observations and free run (black). Horizontal lines exhibit the mean values. Right panels exhibit the histograms of the time-series.

**a) DA forecast for VSL-Min**

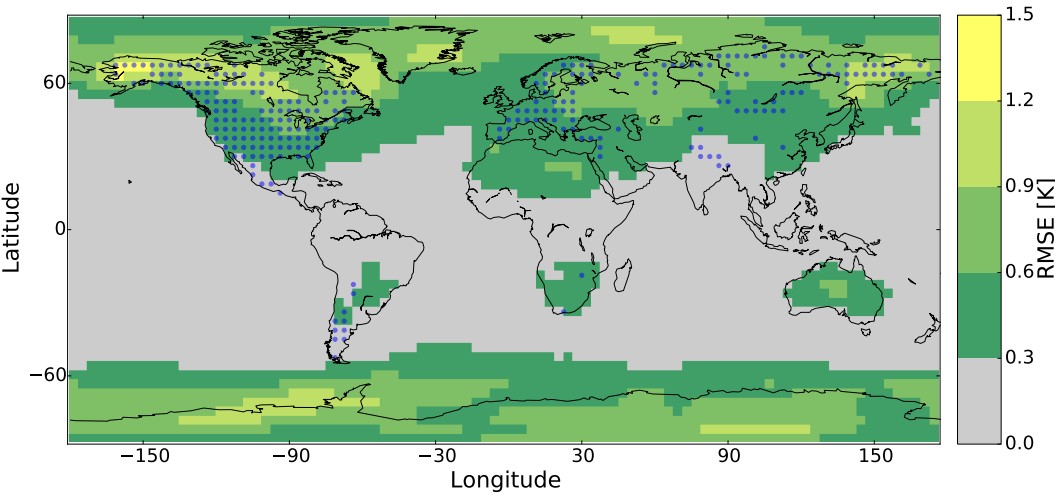

**b) DA analysis for VSL-Min**

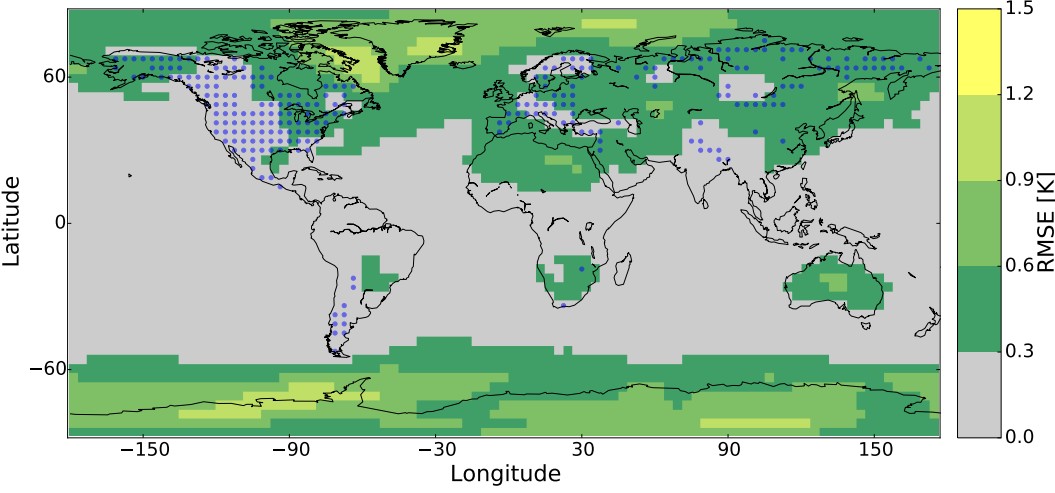

**Figure 7.** Time-averaged RMSEs of SLAB experiment for a) DA forecast and b) DA analysis using the VSL-Min observation operator.

**a) DA forecast for VSL-Prod**

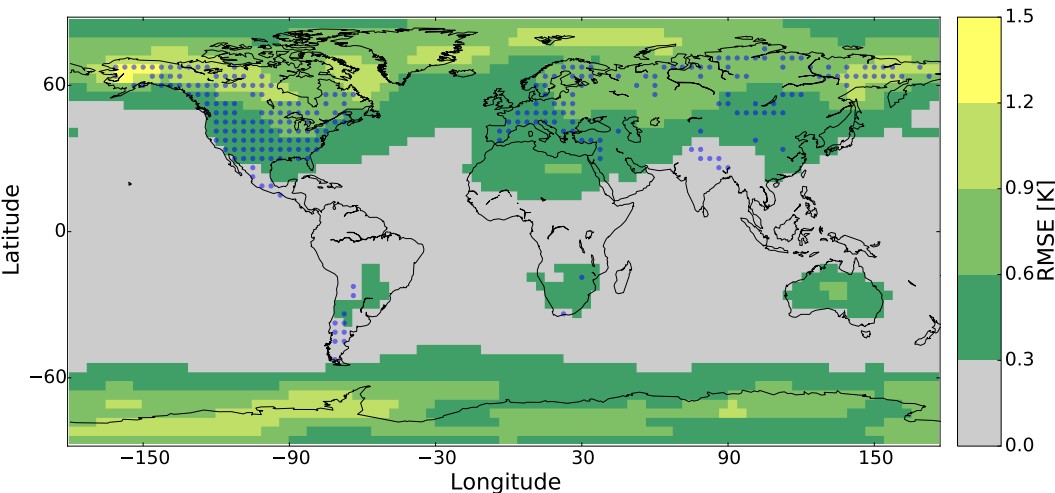

**b) DA analysis for VSL-Prod**

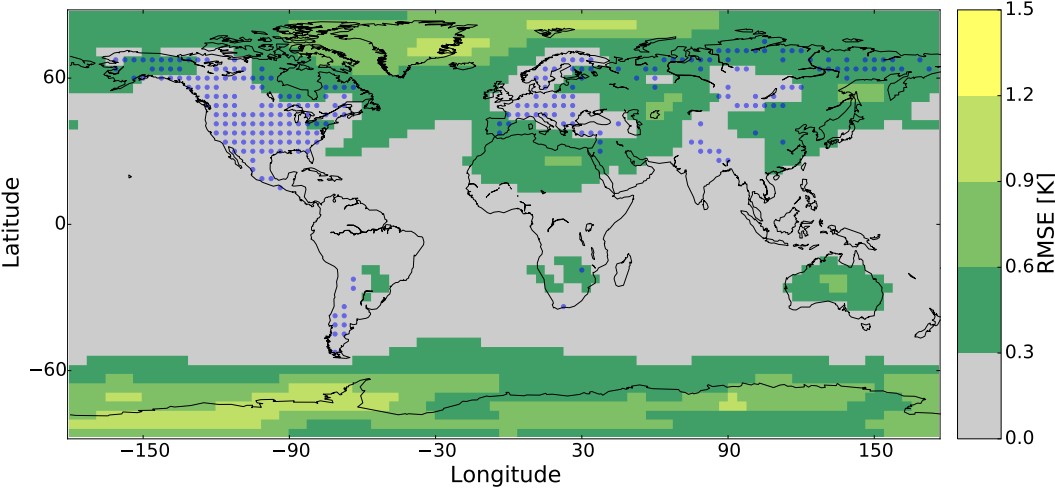

**Figure 8.** Time-averaged RMSEs of SLAB experiment for a) DA forecast and b) DA analysis using the VSL-Prod observation operator.

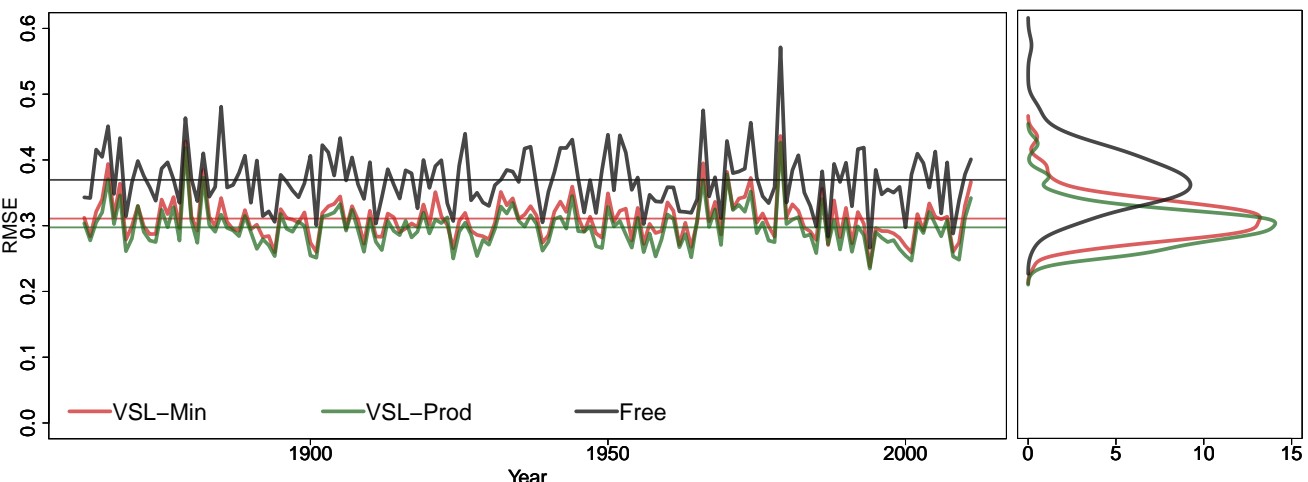

**Figure 9.** Global ensemble mean for analysis constrained by VSL-Min (red) and VSL-Prod (green) pseudo-TRW observations and free run (black). Horizontal lines exhibit the mean values. Right panel exhibits the histograms of the time-series.

**a) DA analysis for VSL-Min with nocycling**

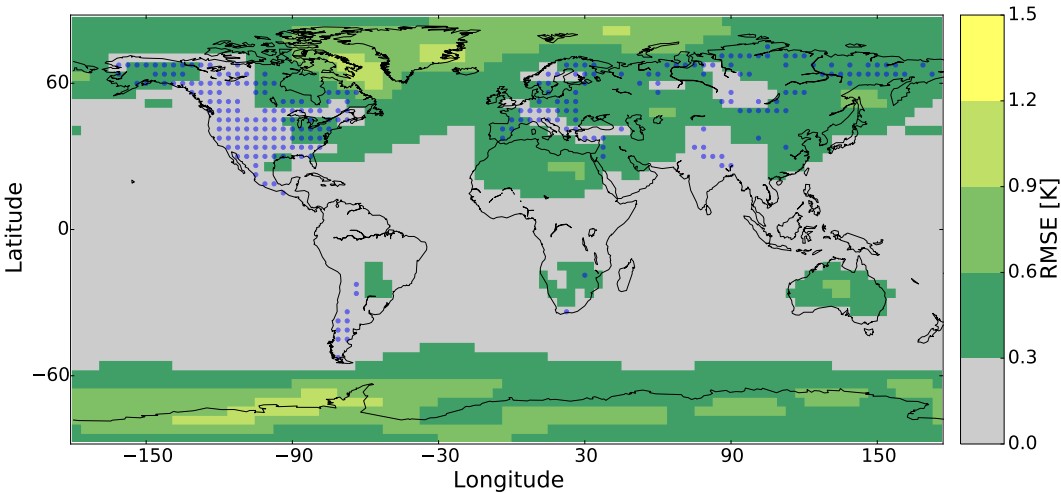

**b) DA analysis for VSL-Prod with nocycling**

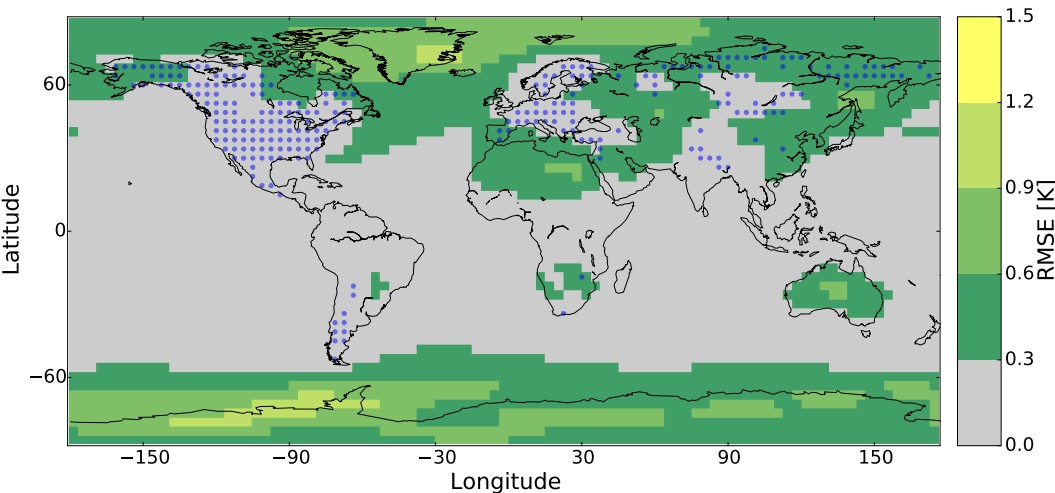

**Figure 10.** Time-averaged RMSEs of SLAB experiment for a) nocycling DA analysis using the VSL-Min and b) the VSL-Prod observation operator.

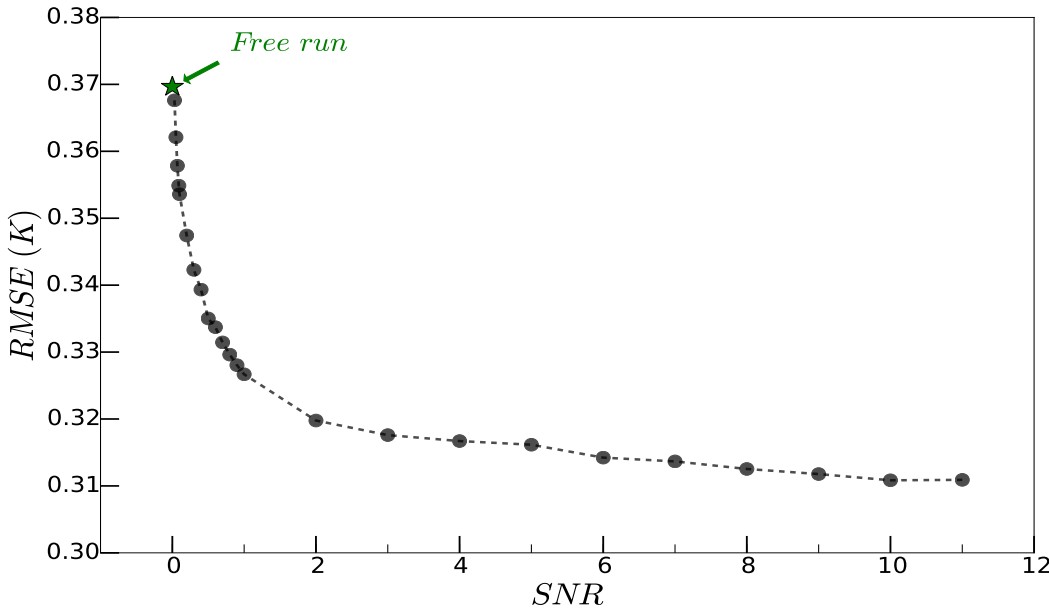

**Figure 11.** Time-averaged global RMSEs of SLAB experiment for nocycling DA using the VSL-Min and different signal to noise ratios. The Green star shows the Free run RMSE.

a)

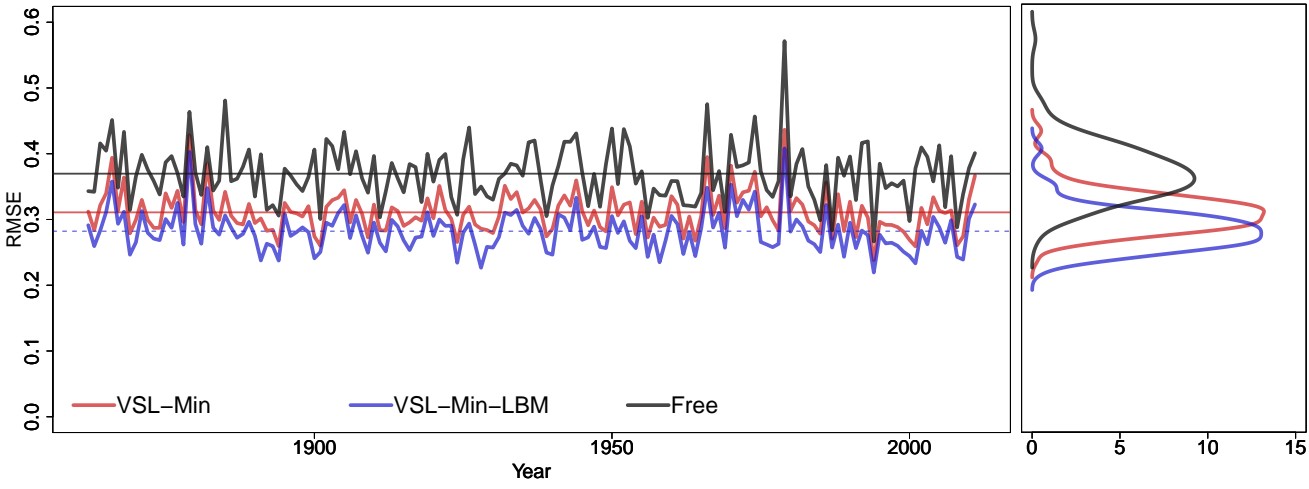

b)

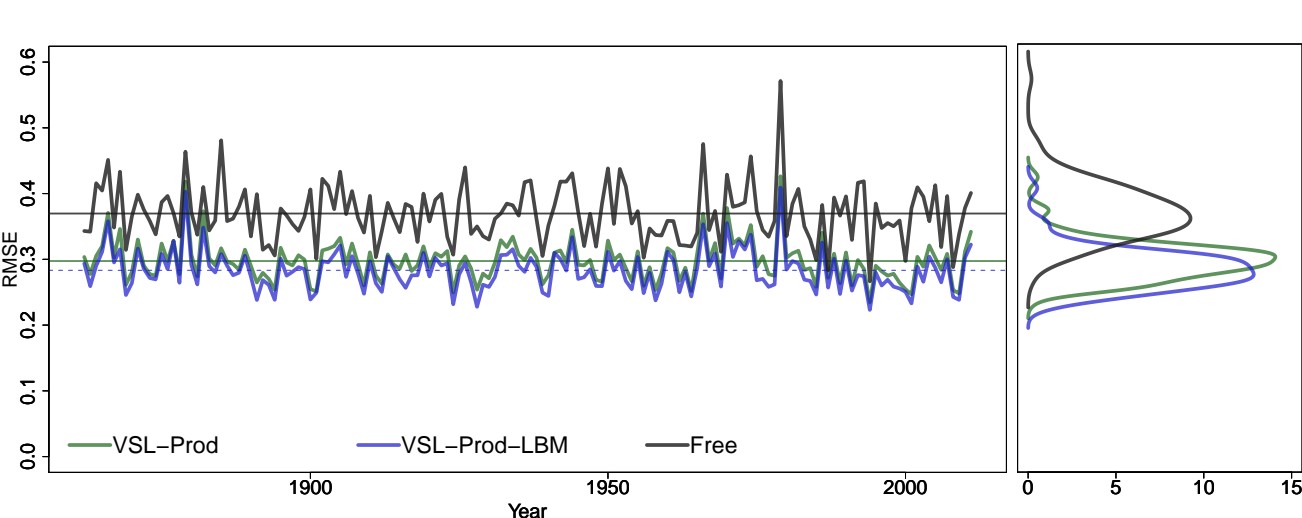

**Figure 12.** Global ensemble mean for analysis constrained by pseudo-TRW observations for a) VSL-Min with the climatological soil moisture (red), with the soil moisture computed by Leaky Bucket Model (blue) and free run (black); b) VSL-Prod with the climatological soil moisture (green), with the soil moisture computed by Leaky Bucket Model (blue) and free run (black). Horizontal lines exhibit the mean values. Right panels exhibit the histograms of the time-series.

**a) DA analysis for VSL-Min with Leaky Bucket Model and nocycling**

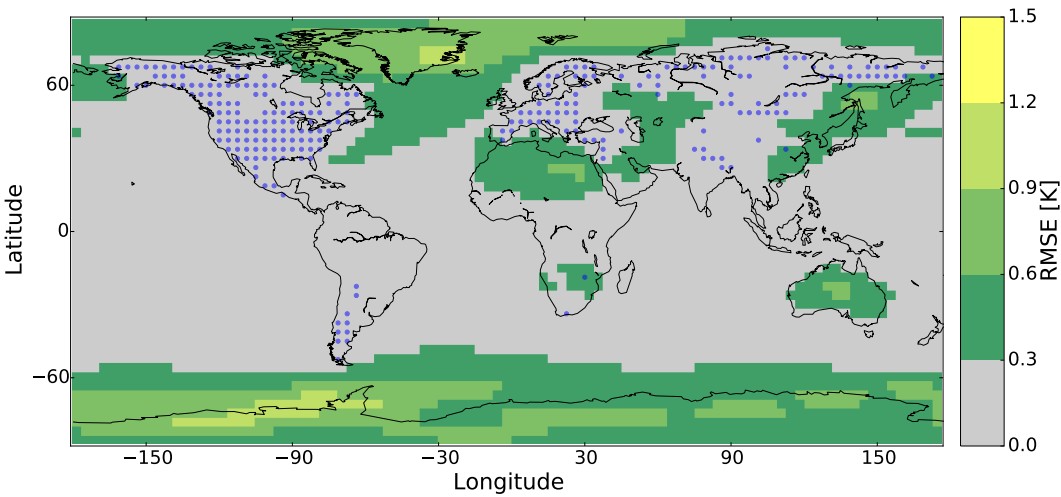

**b) DA analysis for VSL-Prod with Leaky Bucket Model and nocycling**

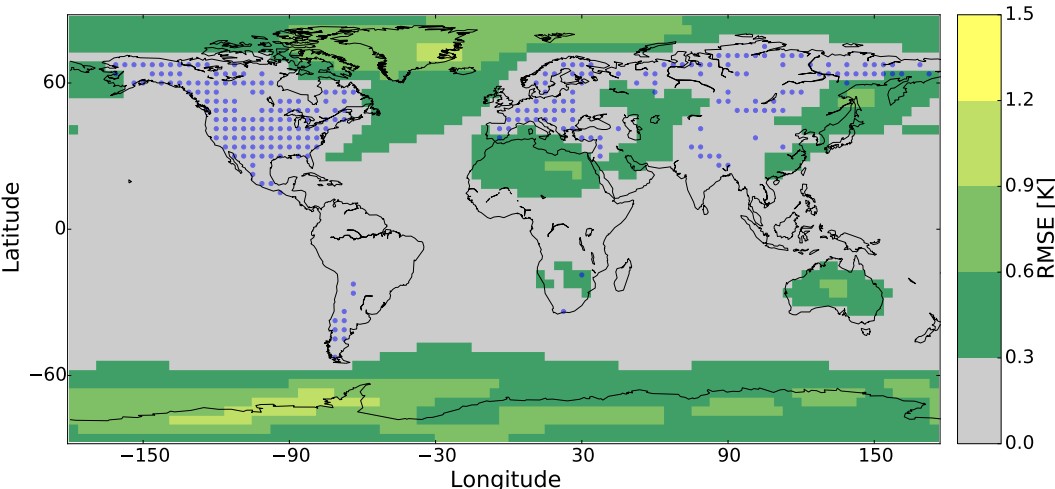

**Figure 13.** Time-averaged RMSEs of SLAB experiment for a) nocycling DA analysis using the VSL-Min with Leaky Bucket Model and b) the VSL-Prod with Leaky Bucket Model observation operator.

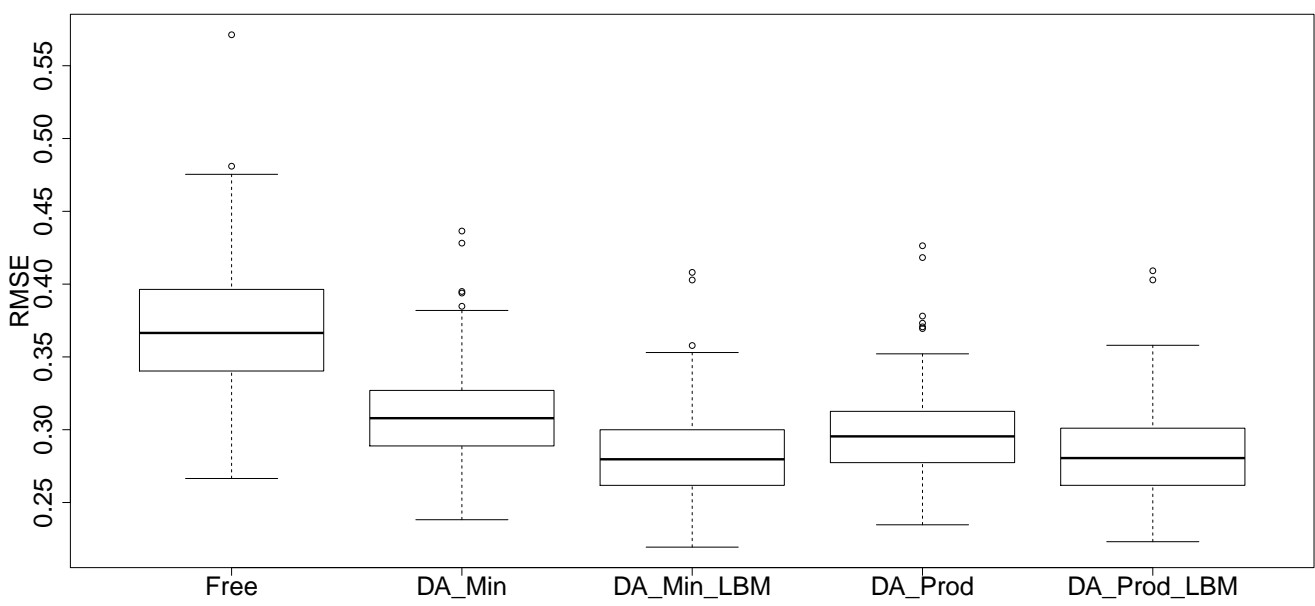

**Figure 14.** Histograms of global ensemble mean for analysis constrained by pseudo-TRW observations for Free run, DA run with VSL-Min, VSL-Prod using the climatological soil moisture and VSL-Min, VSL-Prod using the Leaky Bucket Model (LBM).