# Peer review of "Assimilation of Pseudo-Tree-Ring-Width observations into an Atmospheric General Circulation Model"

_Climate of the Past, 2016_

## Referee Comment (RC1) · Anonymous Referee #1 · 10 Oct 2016

Review for Authors

"Assimilation of Pseudo-Tree-Ring-Width observations into an Atmospheric General Circulation Model"

10/10/16

Summary:

The authors present a nice new study using a version of the IC-GCM SPEEDY adapted for online data assimilation, and test the relative benefits of using a forward model for tree-ring width (VS-Lite) to assimilate proxy observations in a perfect model scenario, and using both offline and online data assimilation methods. The scientific questions

are tested in a piece-wise fashion, and the study finds that the multivariate response of trees may confound their use in data assimilation, but this may be site-specific and related to the climatology; they argue that their method may thus be applied for proxy network and sampling optimization.

Specific Comments:

My major comment for this paper is that while the language is very concise and direct, it is perhaps too concise; many sections of the paper leave the reader wanting more detail or are a bit confusing because they are so brief. For example, in the introduction, I felt as if I didn't have a clear grasp on what new problem the authors were looking to solve, existing gaps in the literature, and how they addressed these in a novel way. There are also a lot of references to previous studies but without any additional information and the reader is left feeling lost. I have made notes in my comments below about the specific sections where more detail is needed. Very few equations describing the experimental design are given to orient the reader to the various components of the DA strategy and how you altered it for your specific set of tests.

There are also some major relevant references that have come before this study that are missing and not discussed, which I have pointed out in my line-by-line comments below. In particular, I think this paper needs to cite and discuss previous findings of Dee, Steiger, Emile-Geay and Hakim 2016 (JAMES):

Dee, Sylvia G., et al. "On the utility of proxy system models for estimating climate states over the common era." Journal of Advances in Modeling Earth Systems (2016).

They have already employed 3 different proxy system models with DA and it would be helpful here to discuss how your study is different from the findings that have recently been outlined in that paper. It's clear that you are performing different tests in this study, but you have to acknowledge that this is not the first piece of work to include forward proxy models with DA (as written you assert this).

From a science perspective, and as I've highlighted below, I think there are some design problems with the VS-Lite application: namely, you used prescribed soil moisture fields when you can instead use time-varying precipitation. Why use a climatological average that is not time-varying for DA when you can use dynamically-updated precip? This makes no sense to me, and I feel it detracts from your results.

In general, with some revisions to the text giving more description, more background, and much more motivation, this paper should be suitable for publication in CoP.

Technical Comments:

Note: The way this manuscript is numbered makes it challenging to give line by line comments. Can you please revise this?

Page 1, Line: 4: appeared = appears (active voice) 13: revise "the so called" to "the usage of paleoclimate proxy records." 14: Revise: "Nonetheless, these natural archives. . ." 16: Revise: "is still an open question" to "can often remain opaque." 17: Delete "To the," rephrase "At present, many. . .. 18: comma after hindcasts, 22: and cite Dee et al., JAMES 2016 in addition to other citations.

Page 2 5: rephrase last sentence "Finally, the use of a particle filter has been tested. . ." 11: cite Dee et al., 2015 (JAMES)—PRYSM along with Evans (review of existing forward models). 12-15: Need to cite and discuss previous findings of Dee, Steiger, Emile-Geay and Hakim 2016 (JAMES) here:

Dee, Sylvia G., et al. "On the utility of proxy system models for estimating climate states over the common era." Journal of Advances in Modeling Earth Systems (2016).

This paper has already employed 3 different proxy system models with DA and it would be helpful here to discuss how your study is different from the findings that have recently been outlined in that paper. It's clear that you are performing different tests in this study, but you have to acknowledge that this is not the first piece of work to include forward proxy models with DA.

17: comma after AC15, delete now, comma after scenario. "were" = "where." 20-25: Back to my major comment above: to a person who is not already quite familiar with the technical details of Data Assimilation, these objectives are opaque. We need more background and you haven't yet defined prior, posterior, etc. There hasn't been any prior introduction of the DA equations so all of this comes out of nowhere. 25: filed = 'field' 28: yes it has already been explicitly investigated, in Dee et al., 2016—as I mentioned, you'll have to discuss this and potentially change the language in your introduction accordingly. 29: rephrase "the TRW forward model, and the climate model.."

Page 3 3: rephrase: "accuracy, relatively user-friendly implementation, and computational expense." 5: what do you mean by "adjoint model"? This is not clear. 9: rephrase "have historically been prohibitively expensive...", hyphen in high-dimensional. Change "However" to "Thus" 10: toy models—is this a common phrase in DA? You have not defined it. I think you should change to "perfect model studies," which is a more widely recognized term for this type of study. Or, "pseudoproxy tests." 13: If I am a person unfamiliar with DA, how do I know what the 'observation operator' is? We could really use some equations here: lay out the DA equations for us so we know what the 'observation operator' is. The DA community will follow you, but most others will not. 13: You have not defined "TA" yet. 15: grammar is incorrect in last part of this sentence. Perhaps you mean: "We study the impact of....using the assimilation of TA linear observations as a reference." 19: what is fuzzy logic? Please cite and explain in detail.

2.2.1 Spell out "V-S-Lite Model" 20: change "limiting factors" to "model inputs for VSL are ...". and put parentheses around (T, M). 21: 'variables' (add s) and rephrase "variables influence tree growth..." delete period after gm, just continue sentence "using a piece-wise" and put colon after Tolwinski citation:

Page 4 1: no indent, no capitalization of Where, change to "denote minimum thresholds for temperature and moisture below which there is no grown, and TU and MU are upper thresholds above which tree growth is optimal" 2: are you sure it's optimal and not too

hot/dry?

2.2.2 the reader does not know what Fuzzy Logic is because you have not introduced it in the text, nor have you cited it. We need more context. 9: delete 'have, the' . . . and what is PLF? Have you defined this yet? 10-15: this is too brief and we need more motivation here about your experimental design and what you're testing.

Page 5 1: delete 'the' before version 32 1-7: be a bit careful here with text—this reads awfully similar to Molteni 2003 6: rephrase "The latter makes SPEEDY…" 7: change 'presented in this paper' to 'necessary for this study' 15-16: not enough information for a non-DA specialist. 21: 'where' = were 23: huge = large, change 'are' to 'were' – also, what is the fallout of this? 24: change 'as the following' to 'as follows:' 25: delete comma after deviation 2.3.3. change to 'Simulations' instead of Runs characteristics, which is not grammatically correct.

Page 6, Line:

2 consist = 'consists' 6 'from the equilibrium' —not enough detail. Do you mean it's already spun up, or it's a control simulation? Be more clear. 7 should not be a new paragraph, change 'affordability' to 'efficiency' 8 "minimum" and "product" Triangular norms come out of no where, we need an explanation, description, citation, and to not be lost by the first use of these terms 10 150 year (no 's') — and, by 'nature' run do you mean 'control' run? I have never seen the term 'nature' run. change wording. 11 change month to 'months' 12 nature = control, when you say 'different ensemble runs' are these the ensemble of climate state vectors or 'prior' for the DA? be clear. Change 'driving' to 'forcing SPEEDY' 16 change 'added to the clean' to 'imposed on the TA observations' — also I think you should spell out TA and not abbreviate. It's a short acronym and it's confusing when there are already so many other acronyms flying around. Delete comma after 'observations so as to obtain'.. 17 10 seems like a very high and unrealistic SNR for a pseudo proxy test. See previous literature on this topic, and the Smerdon et al. 2012 review. 20 So, this seems very unsatisfying.

Even though SPEEDY has a climatological mean soil moisture field, precipitation, by contrast, is varying. You can run VSLite with Precipitation and a parameterization that goes from precip to soil moisture—people run VS Lite this way all the time, and I don't think it make sense not to in this case. I would redo all the pseudo proxy analysis with time-varying precipitation instead of time-invariant climatological mean soil moisture. . ..I have seen this mistake before with VSLite and it causes an unphysical response for the trees. 30 citation needed after 'internal variability'

Page 7, Line: 1 rephrase to "Our results are presented in three sections: 1). . ." 2-4 This is confusing—what do you mean by the word 'selection'? Elaborate. Add 'the' before 'temperature' 8 rephrase "disentangled to some extent by considering atmospheric variability to be a superposition.." 24 change but present. . . to "stationary and fluctuate over longer time scales. These low-frequency". . . 25 occur should be 'occurs' 26 reverse order of wording to read 'modes of variability' 27 which annular modes? this is a very offhand reference. 28 change to 'displacements of the jet stream' 29 no comma after SPEEDY, nature=control

Page 8, Line: 2 again I think nature should be control throughout. Larger comment for Section 3.2.1: We need more information on the experimental design—perhaps a graphic showing a schematic of your experimental design and the PSM vs. no PSM simulations, online vs. offline, showing the full scope of the research you performed for this paper. What is the point of the control run in this context? It's just not very clear in the current text. How did you use it?

15 change 'there exists a DA skill' — awkward wording, revise for clarity 18 rephrase to 'proxy record locations', and the comma after Northern hemisphere should be a semi-colon (;) 20 no comma after 'skill' 24 rephrase: "constrain temperature with considerably larger skill than TRW sites in South Africa. This finding may prove useful for the design of optimal TRW chronology networks. . .." 25 you need to cite Comboul et al., 2015 here which is also about optimizing observing networks in paleoclimate data, and discuss their findings (using coral pseudo proxies) in relation to yours: "

CITATION: Comboul, Maud, et al. "Paleoclimate Sampling as a Sensor Placement Problem." Journal of Climate 28.19 (2015): 7717-7740.

29 citations are out of chronological order, change last bit of sentence from 'is currently' to 'is generally termed 'offline Data Assimilation.' 30 rephrase end 'using assimilation, the prior. . .'

Page 9, Line: 1 can you remind the reader about the differences between the two pseudo proxy schemes here ? MIN vs PROD? Give us a brief description to re-orient, as well as your hypothesis for how the two will differ. 5 delete comma after VSL-Min, add 'as a TRW observation . . .' 6 rephrase "analysis, as demonstrated in Figure 6b. The expected value of the RMSE shifts significantly toward lower values.." 8 change present to 'shows' 10 revise "performs with slightly better skill"

Note: there's no discussion of the pseudo proxy design here. . ... 17 What is TA DA???? Just write it out. 18 change to 'applied in parallel and independently of any specific. . ." 25-30 again cite Comboul et al., here:

Comboul, Maud, et al. "Paleoclimate Sampling as a Sensor Placement Problem." Journal of Climate 28.19 (2015): 7717-7740.

—there is an official term for this kind of work, and it's optimal sensor placement (OSP)—much literature here in the pseudo proxy community and forward modeling/proxy system modeling that you need to work through in this discussion. Also, be careful with your language here. . ...is this really a fair statement to make when you didn't use time-variant soil moisture? if you are going to make the claim that your method can be used to design OSSEs you should probably give a walk-through, thorough example of this and associated caveats. Show a map of where the trees capture the most climate variability, etc. Also, the claim that you can apply this method to any proxy with 'stable time resolution' needs to be clarified. Do you mean annual resolution? It would be difficult to do this with lower frequency climate data like sediment cores or speleothems. So, this comment seems a bit far-reaching.

Page 10, Line:

5 delete comma after provided, delete 'the' before results, delete "huge amount of" 6-7 revise language for clarity—'undiscriminated' — I think you mean 'indiscrimi- nate' ? 9 change 'In addition to the classical DA approaches used in paleoclimate studies..." 10 and cite Dee et al., 2016 as well, which also uses this approach AND PSMs... 18 change "In this conditions" which is grammatically incorrect to "Under these conditions..." 19 what is meant by 'climatological levels?' 20 delete 'model' after SPEEDY, and the phrase "it is not surprising to enter the offline ..." is confusing and needs to be revised for clarity 22 delete "In this state of affairs" and change to Thus, it seems unlikely ... 23 constraint = constraints 24 this is too brief and we need examples—of course there is climate variability on time scales longer than 1 year. The obvious one is ENSO, but you need to give more examples and more citations. 25 rephrase "Accordingly, we expect that it should be possible to obtain...." and change 'skills' to skill. 27 rephrase "It is not clear if whether we can employ this technique with SPEEDY to properly estimate..." 28 comma after In particular,

Page 11, Line: 4 rephrase "conducted with SPEEDY support results obtained..." 9 delete colon (:) 10 rephrase "contained in them and the..." 11-15 it's not clear from the current text what point you're making here. Revise for clarity.

18 'response saturation'—what is this? The paper is jargon-y, as I mentioned. We need more description of these terms. 22 be careful here.....VSLite is not very Gaus- sian either. There is a brief discussion of this in Dee et al., 2016 in the TRW sec- tion. What is the fall out of this? General: we need a concrete summary of your findings—there isn't a conclusion section that gives us a summary and broader im- plications of your work. Needs to be added.

Page 12 Appendix—you need to spell out the meaning of OSSE on first use—can- not abbreviate.

Please also note the supplement to this comment:
http://www.clim-past-discuss.net/cp-2016-92/cp-2016-92-RC1-supplement.pdf

---

## Referee Comment (RC2) · Anonymous Referee #2 · 24 Oct 2016

Review of "Assimilation of Pseudo-Tree-Ring-Width observations into an Atmospheric General Circulation Model", by Walter Acevedo1 Bijan Fallah, Sebastian Reich, and Ulrich Cubasch

This is a nice study on the assimilation of tree ring width into an atmospheric GCM. It systematically tests several open questions that are relevant to the community using a very simple climate model and assimilation set-up. The paper is well written, though it caters the specialist more than the general reader. It fits very well within the scope of the journal. I recommend publication after some revisions are taken into account.

One important point which I feel is not treated adequately in this paper is the observation error. The authors use a signal-to-noise ratio of 10. Typical pseudoproxy experiments use ratios of 0.25-1. Although the authors mention in the last part of their paper that their signal-to-noise ratio is optimistic, the reader is left wondering what the effect could be. Furthermore, the error model is not well explained. Why white noise? What would be the effect of a spatial error structure? What would be the effect of systematic spectral biases in the tree rings? Even more importantly: Was the error assumed to be known perfectly? These questions would be very important for the community and would probably deserve a dedicated paper, but to the extent to which they could interfere with some of the results presented, I think some discussion should be added.

A second point concerns the model description, which is rather short. In particular, the boundary conditions are not well discussed (e.g., greenhouse gases, volcanic aerosols, etc.). I am aware that this is a Observation System Simulation Experiment, nevertheless I would be interested in the effects of boundary conditions. What are the climatological maps from ECMF used for? And maps of what quantities? The paper is sufficiently short; some more explanations could be added here.

The authors use many acronyms (TRW, PLF, DA, SNR, VSL, GCM, TA, EnKF, CFR, OSSE) which may be familiar to some readers but not to others. Again, I don't think that the paper is too long, and some of the acronyms could be spelled out for the sake of better readability.

The description not only of the methods, but also of the result is rather short.

p. 3, l. 6: Or, covariance matrices may be blended from the ensemble and other estimations.

p. 4, l. 10: Explain t-norms.

p. 6, l. 28: "a fixed averaging period length of one year": How was that year defined? April to March?

p. 7: The reader might get confused with the terms "run" (nature run, free ensemble run) and experiment (PRESCRIBED, SLAB). The table does not help the confusion,

but the Appendix does, it is very well written. Please refer at the appropriate places in the manuscript to the Appendix.

p. 7, l. 21: The low yearly internal variability in the tropics deserves some further attention. What does this mean in relation to real-world phenomena such as ENSO or PDO? This is particularly interesting as the authors discuss the PRESCRIBED set up and the SLAB but later note that fully coupled systems could/should be used. Would the result be completely different in the tropics?

p. 7, l. 25: Just really minor: "Fig. 3a" is arguably more common than "figure 3.a"

p. 8, l.16 and elsewhere: Is the emphasis (bold italics) necessary? The authors use the term in the same way as the literature.

p. 9, l. 18: What do you mean with "any specific year"? Does that mean that the boundary conditions are disregarded? Can 1900 serve as a prior for 1999?

p. 10, l. 13: "a more consistent"?

p. 10, l. 30: There is another important difference to traditional CFR techniques (by the way: spell out), namely that data assimilation at least formally does not require calibration and thus is less sensitive to stationarity issues.

p. 11, l. 1: "full atmosphere-ocean interaction".

p. 11, l. 25: Not only model errors, also the observation error is an issue.
* * *

---

## Author Response (AR1)

**Final answer to the Comments**

Acevedo et al.,

January 17, 2017

Dear Prof. Goosse and Dear reviewers,

Thank you so much for your constructive comments. A track changes version of the manuscript is also added at the end of this answer(Red is deleted and blue is added). We reply to all your comments here:

**1 Answer to Reviewer 1**

We wish to thank you so much for your constructive review and very detailed comments. It would be our pleasure to do all the modifications and make the improvements you have suggested, in the next version of the manuscript. We answer your comments (*italic*) point by point (**Bold**):

*My major comment for this paper is that while the language is very concise and direct, it is perhaps too concise; many sections of the paper leave the reader wanting more detail or are a bit confusing because they are so brief. For example, in the introduction, I felt as if I didn't have a clear grasp on what new problem the authors were looking to solve, existing gaps in the literature, and how they addressed these in a novel way.*
*There are also a lot of references to previous studies but without any additional infor- mation and the reader is left feeling lost. I have made notes in my comments below about the specific sections where more detail is needed. Very few equations describing the experimental design are given to orient the reader to the various components of the DA strategy and how you altered it for your specific set of tests. There are also some major relevant references that have come before this study that are missing and not discussed, which I have pointed out in my line-by-line comments below.*

**We completely agree. Therefore, we have reviewed the methods in more detail and added additional text to the introduction and methods as well as the results and the discussions (We expanded the manuscript from 26 pages to 35 pages). Data assimilation equations are also added in the methods. Subsections "Rationale, Kalman Filter, Ensemble Kalman Filter, Time Averaged Ensemble kalman Filter, OSSE, TRW forward models and VSL from the Fuzzy Logic Viewpoint" are fully explained and added to the new version. We**

hope that the new version of the manuscript with all additional subsections and explanations is now clear for the readers.

*In particular, I think this paper needs to cite and discuss previous findings of Dee, Steiger, Emile-Geay and Hakim 2016 (JAMES):*

*Dee, Sylvia G., et al. "On the utility of proxy system models for estimating climate states over the common era." Journal of Advances in Modeling Earth Systems (2016).*

*They have already employed 3 different proxy system models with DA and it would be helpful here to discuss how your study is different from the findings that have recently been outlined in that paper. It's clear that you are performing different tests in this study, but you have to acknowledge that this is not the first piece of work to include forward proxy models with DA (as written you assert this).*

**We agree on this and the point that it is not the first study considering the forward proxy models with DA. We have also cited the paper by [Dee et al., 2016] in the new version and modified the Introduction accordingly, acknowledging this study. Given that this novel paper [Dee et al., 2016] was published first on $10^{th}$ August 2016, we missed that during the production process of our manuscript. This paper is a perfect reference and back-up for our similar strategy in paleoclimate reanalysis.**

*From a science perspective, and as I've highlighted below, I think there are some de- sign problems with the VS-Lite application: namely, you used prescribed soil moisture fields when you can instead use time-varying precipitation. Why use a climatological average that is not time-varying for DA when you can use dynamically-updated precip? This makes no sense to me, and I feel it detracts from your results.*

**The soil moisture used in our experiments is not just a climatological average but a monthly climatology, meaning that the soil moisture value does depend on the month and is therefore time varying, allowing the seasonal competition of limiting factors to develop. Using SPEEDY precipitation as input for VS-Lite model is indeed a possibility to add time variability to the soil moisture, however this would imply to use the CPC Leaky Bucket [Huang et al., 1996a] Model in order to generate moisture time series out of temperature and precipitation time series. We have implemented the Leaky Bucket Model and repeated two simulations. Please see the answer to the Editor or Figures 12-14 and subsection 3.2.4 of the new manuscript.**

*In general, with some revisions to the text giving more description, more background, and much more motivation, this paper should be suitable for publication in CoP. Note: The way this manuscript is numbered makes it challenging to give line by line comments. Can you please revise this?*

**Thanks for the comment. We have used the CoP's LATEX tem-**

**plate for compiling the manuscript. Maybe this issue can be suggested to the Editorial Support, providing a new template for CoP.**

*Page 1, Line: 4: appeared = appears (active voice) 13: revise "the so called" to "the usage of paleoclimate proxy records." 14: Revise: "Nonetheless, these natural archives. . ." 16: Revise: "is still an open question" to "can often remain opaque." 17: Delete "To the," rephrase "At present, many. . .. 18: comma after hindcasts, 22: and cite Dee et al., JAMES 2016 in addition to other citations.*

**Done!**

*Page 2 5: rephrase last sentence "Finally, the use of a particle filter has been tested..." 11: cite Dee et al., 2015 (JAMES)—PRYSM along with Evans (review of existing forward models). 12-15: Need to cite and discuss previous findings of Dee, Steiger, Emile-Geay and Hakim 2016 (JAMES) here: Dee, Sylvia G., et al. "On the utility of proxy system models for estimating climate states over the common era." Journal of Advances in Modeling Earth Systems (2016).*
**Done!**
*17: comma after AC15, delete now, comma after scenario. "were" = "where." 20-25: Back to my major comment above: to a person who is not already quite familiar with the technical details of Data Assimilation, these objectives are opaque. We need more background and you haven't yet defined prior, posterior, etc. There hasn't been any prior introduction of the DA equations so all of this comes out of nowhere.*

**Done!**
*25: filed = 'field' 28: yes it has already been explicitly investigated, in Dee et al., 2016—as I mentioned, you'll have to discuss this and potentially change the language in your introduction accordingly. 29: rephrase "the TRW forward model, and the climate model.."*

**We have deleted this sentence and acknowledged Dee et al., 2016.**

*Page 3 3: rephrase: "accuracy, relatively user-friendly implementation, and computational expense." 5: what do you mean by "adjoint model"? This is not clear. 9: rephrase "have historically been prohibitively expensive...", hyphen in high- dimensional. Change "However" to "Thus" 10: toy models—is this a common phrase in DA? You have not defined it. I think you should change to "perfect model studies," which is a more widely recognized term for this type of study. Or, "pseudoproxy tests."*

**We changed the text accordingly.**

*13: If I am a person unfamiliar with DA, how do I know what the 'observation operator' is? We could really use some equations here: lay out the DA equations for us so we know what the 'observation operator' is. The DA community will follow you, but most others will not. 13: You have not defined "TA" yet.*

**Several sections ans subsections have been added to the manuscript to cover your comment. We also reordered many parts of the manuscript to describe every detail of the methodology.**

*15: grammar is incorrect in last part of this sentence. Perhaps you mean: "We study the impact of....using the assimilation of TA linear observations as a reference." 19: what is fuzzy logic? Please cite and explain in detail. 2.2.1 Spell out "V-S-Lite Model" 20: change "limiting factors" to "model inputs for VSL are ..." and put parentheses around (T, M). 21: 'variables' (add s) and rephrase "variables influence tree growth..." delete period after gm, just continue sentence "using a piece-wise" and put colon after Tolwinski citation:*

**Done!**

*Page 4 1: no indent, no capitalization of Where, change to "denote minimum thresholds for temperature and moisture below which there is no grown, and TU and MU are upper thresholds above which tree growth is optimal" 2: are you sure it's optimal and not too hot/dry?*

**Done! TU and MU are defined as optimal growth limits and not the hottest or driest limits.**

*2.2.2 the reader does not know what Fuzzy Logic is because you have not introduced it in the text, nor have you cited it. We need more context. 9: delete 'have, the' ... and what is PLF? Have you defined this yet? 10-15: this is too brief and we need more motivation here about your experimental design and what you're testing.*

**We added a complete section describing the Fuzzy Logic concept.**

*Page 5 1: delete 'the' before version 32 1-7: be a bit careful here with text—this reads awfully similar to Molteni 2003 6: rephrase "The latter makes SPEEDY..." 7: change 'presented in this paper' to 'necessary for this study' 15-16: not enough information for a non-DA specialist. 21: 'where' = were 23: huge = large, change 'are' to 'were' – also, what is the fallout of this? 24: change 'as the following' to 'as follows:' 25: delete comma after deviation 2.3.3. change to 'Simulations' instead of Runs characteristics, which is not grammatically correct.*

**Done. We think that the new section added will cover some basics of the DA implementation.**

*Page 6, Line: 2 consist = 'consists' 6 'from the equilibrium' —not enough detail. Do you mean it's already spun up, or it's a control simulation? Be more clear. 7 should not be a new paragraph, change 'affordability' to 'efficiency' 8 "minimum" and "product" Triangular norms come out of no where, we need an explanation, description, citation, and to not be lost by the first use of these terms 10 150 year (no 's') — and, by 'nature' run do you mean 'control' run? I have never seen the term 'nature' run. change wording. 11 change month to 'months' 12 nature = control, when you say 'different ensemble runs' are these*

*the ensemble of climate state vectors or 'prior' for the DA? be clear. Change 'driving' to 'forcing SPEEDY' 16 change 'added to the clean' to 'imposed on the TA observations' — also I think you should spell out TA and not abbreviate. It's a short acronym and it's confusing when there are already so many other acronyms flying around. Delete comma after 'observations so as to obtain'..*

**Done**

*17 10 seems like a very high and unrealistic SNR for a pseudo proxy test. See previous literature on this topic, and the Smerdon et al. 2012 review.*

**Thanks a lot for this comment. We have redone the offline DA using VSL-min for 24 different SNR values (from $SNR = $ 0.03 to 11) and plotted the time-averaged global RMSEs. We have added Figure 11 and a subsection for SNR to the manuscript. As can be seen the plot shows an elbow around value $SNR = 1$ and reaches the Free run at around $SNR = 0.03$ where almost all of the observations are neglected in DA..**

*20 So, this seems very unsatisfying. Even though SPEEDY has a climatological mean soil moisture field, precipitation, by contrast, is varying. You can run VSLite with Precipitation and a parameterization that goes from precip to soil moisture—people run VS Lite this way all the time, and I don't think it make sense not to in this case. I would redo all the pseudo proxy analysis with time-varying precipitation instead of time- invariant climatological mean soil moisture….I have seen this mistake before with VSLite and it causes an unphysical response for the trees.*

**We already answered this on page 2 of this answer.**

*30 citation needed after 'internal variability'*

**Done!**

*Page 7, Line: 1 rephrase to "Our results are presented in three sections: 1)…" 2-4 This is confusing—what do you mean by the word 'selection'? Elaborate. Add 'the' before 'temperature' 8 rephrase "disentangled to some extent by considering atmospheric variability to be a superposition.." 24 change but present… to "stationary and fluctuate over longer time scales. These low-frequency"… 25 occur should be 'occurs' 26 reverse order of wording to read 'modes of variability' 27 which annular modes? this is a very offhand reference. 28 change to 'displacements of the jet stream' 29 no comma after SPEEDY, nature=control*

**Done.**

*Page 8, Line: 2 again I think nature should be control throughout. Larger comment for Section 3.2.1: We need more information on the experimental design—perhaps a graphic showing a schematic of your experimental design and the PSM vs. no PSM simulations, online vs. offline, showing the full scope of*

*the research you performed for this paper. What is the point of the control run in this context? It's just not very clear in the current text. How did you use it?*

**Figure 1 is illustrating the schematic of our experiment. Here we refer to the figure 1 in the new manuscript and import the Appendix in the main part of the paper along with figure 1.**

*15 change 'there exists a DA skill' — awkward wording, revise for clarity 18 rephrase to 'proxy record locations', and the comma after Northern hemisphere should be a semi-colon (;) 20 no comma after 'skill' 24 rephrase: "constrain temperature with considerably larger skill than TRW sites in South Africa. This finding may prove useful for the design of optimal TRW chronology networks...." 25 you need to cite Comboul et al., 2015 here which is also about optimizing observing networks in paleoclimate data, and discuss their findings (using coral pseudo proxies) in relation to yours: " CITATION: Comboul, Maud, et al. "Paleoclimate Sampling as a Sensor Placement Problem." Journal of Climate 28.19 (2015): 7717-7740. 29 citations are out of chronological order, change last bit of sentence from 'is currently' to 'is generally termed 'offline Data Assimilation.' 30 rephrase end 'using assimilation, the prior...'*

**Done.**

*Page 9, Line: 1 can you remind the reader about the differences between the two pseudo proxy schemes here ? MIN vs PROD? Give us a brief description to re-orient, as well as your hypothesis for how the two will differ. 5 delete comma after VSL-Min, add 'as a TRW observation ...' 6 rephrase "analysis, as demonstrated in Figure 6b. The expected value of the RMSE shifts significantly toward lower values.." 8 change present to 'shows' 10 revise "performs with slightly better skill" Note: there's no discussion of the pseudo proxy design here..... 17 What is TA DA???? Just write it out. 18 change to 'applied in parallel and independently of any specific..." 25-30 again cite Comboul et al., here: Comboul, Maud, et al. "Paleoclimate Sampling as a Sensor Placement Problem." Journal of Climate 28.19 (2015): 7717 7740.*

**Done.**

*—there is an official term for this kind of work, and it's optimal sensor placement (OSP)—much literature here in the pseudo proxy community and forward modeling/proxy system modeling that you need to work through in this discussion. Also, be careful with your language here.....is this really a fair statement to make when you didn't use time-variant soil moisture? if you are going to make the claim that your method can be used to design OSSEs you should probably give a walk-through, thorough example of this and associated caveats. Show a map of where the trees capture the most climate variability, etc. Also, the claim that you can apply this method to any proxy with 'stable time resolution' needs to be clarified. Do you mean annual resolution? It would be difficult to do this with lower frequency climate data like sediment cores or speleothems. So, this comment seems a bit far-reaching.*

**We have explained that in the new version of the manuscript.**

We cited the OSP [Ancell and Hakim, 2007, Hakim and Torn, 2008, Mauger et al., 2013, Comboul et al., 2015] here with a discussion on the caveats.

*Page 10, Line: 5 delete comma after provided, delete 'the' before results, delete "huge amount of" 6-7 revise language for clarity—'undiscriminated' — I think you mean 'indiscriminate' ? 9 change 'In addition to the classical DA approaches used in paleoclimate studies..." 10 and cite Dee et al., 2016 as well, which also uses this approach AND PSMs... 18 change "In this conditions" which is grammatically incorrect to "Under these conditions..." 19 what is meant by 'climatological levels?' 20 delete 'model' after SPEEDY, and the phrase "it is not surprising to enter the offline ..." is confusing and needs to be revised for clarity 22 delete "In this state of affairs" and change to Thus, it seems unlikely ... 23 constraint = constraints 24 this is too brief and we need examples—of course there is climate variability on time scales longer than 1 year. The obvious one is ENSO, but you need to give more examples and more citations. 25 rephrase "Accordingly, we expect that it should be possible to obtain...." and change 'skills' to skill. 27 rephrase "It is not clear if whether we can employ this technique with SPEEDY to properly estimate..." 28 comma after In particular,*

**Done.**

*Page 11, Line: 4 rephrase "conducted with SPEEDY support results obtained..." 9 delete colon (:) 10 rephrase "contained in them and the..." 11-15 it's not clear from the current text what point you're making here. Revise for clarity. 18 'response saturation'—what is this? The paper is jargon-y, as I mentioned. We need more description of these terms. 22 be careful here.....VSLite is not very Gaussian either. There is a brief discussion of this in Dee et al., 2016 in the TRW section. What is the fall out of this? General: we need a concrete summary of your findings—there isn't a conclusion section that gives us a summary and broader implications of your work. Needs to be added.*

**The comments are considered in the new manuscript. "response saturation" is changed to "the threshold, for temperature or moisture, after which the growth response does not change". Non-Gaussianity of the VSL is also a challenge for EnKF. A complete subsection have been added at the end of the manuscript to cover a summary and broader implications of our work.**

*Page 12 Appendix—you need to spell out the meaning of OSSE on first use—cannot abbreviate.*
**We moved the appendix to the main text after figure 1.**

**2 Answer to Reviewer 2**

We wish to thank you so much for your positive review and constructive comments. We answer your comments (*italic*) point by point (**Bold**):

*One important point which I feel is not treated adequately in this paper is*

*the observa- tion error. The authors use a signal-to-noise ratio of 10. Typical pseudoproxy experiments use ratios of 0.25-1. Although the authors mention in the last part of their paper that their signal-to-noise ratio is optimistic, the reader is left wondering what the effect could be.*

**This has been also asked by the first reviewer and new Figure 11 is explaining this issue. A block of text also is added to the new version of manuscript explaining this issue.**

*Furthermore, the error model is not well explained. Why white noise? What would be the effect of a spatial error structure? What would be the effect of system- atic spectral biases in the tree rings? Even more importantly: Was the error assumed to be known perfectly? These questions would be very important for the community and would probably deserve a dedicated paper, but to the extent to which they could interfere with some of the results presented, I think some discussion should be added.*

**The signal to noise ratio (SNR) is expressed as the ratio of the standard deviation of the nature (true) run time-series to that of the additive white noise. The measurements' error is assumed not to be correlated in time (no memory), therefore the white noise is usually used in such studies (for example see [Dee et al., 2016] or [McShane and Wyner, 2011] ). Some pieces of text are added to explain the model's error in subsection Observation generation.**

*A second point concerns the model description, which is rather short. In particular, the boundary conditions are not well discussed (e.g., greenhouse gases, volcanic aerosols, etc.). I am aware that this is a Observation System Simulation Experiment, nevertheless I would be interested in the effects of boundary conditions. What are the climatological maps from ECMF used for? And maps of what quantities? The paper is sufficiently short; some more explanations could be added here.*

**Done. We added some explanations about the boundary conditions.**

*The authors use many acronyms (TRW, PLF, DA, SNR, VSL, GCM, TA, EnKF, CFR, OSSE) which may be familiar to some readers but not to others. Again, I don't think that the paper is too long, and some of the acronyms could be spelled out for the sake of better readability.*

**We agree. We expanded the manuscript from 26 pages to 32 pages and spelled out many of the abbreviations.**

*The description not only of the methods, but also of the result is rather short.*

**We added additional text to the discussions as well as methods.**

*p. 3, l. 6: Or, covariance matrices may be blended from the ensemble and other estimations.*

**Done.**

*p. 4, l. 10: Explain t-norms.*

**It is fully explained now in the new manuscript.**

*p. 6, l. 28: "a fixed averaging period length of one year": How was that year defined? April to March?*

**Has been changed to "Given the annual resolution of TRW chronologies, we study the filter performance for yearly averaged values (near surface temperatures)."**

*p. 7: The reader might get confused with the terms "run" (nature run, free ensemble Discussion paper run) and experiment (PRESCRIBED, SLAB). The table does not help the confusion, but the Appendix does, it is very well written. Please refer at the appropriate places in the manuscript to the Appendix.*

**Appendix is moved after figure 1 and the OSSE is described there fully.**

*p. 7, l. 21: The low yearly internal variability in the tropics deserves some further attention. What does this mean in relation to real-world phenomena such as ENSO or PDO? This is particularly interesting as the authors discuss the PRESCRIBED set up and the SLAB but later note that fully coupled systems could/should be used. Would the result be completely different in the tropics?*

**This issue was raised also by reviewer 1. So we discussed this issue in the new subsection Outlook. Giving more examples of phenomena with larger time-scales than one year.**

*p. 7, l. 25: Just really minor: "Fig. 3a" is arguably more common than "figure 3.a"*
**Done.**

*p. 8, l.16 and elsewhere: Is the emphasis (bold italics) necessary? The authors use the term in the same way as the literature.*

**Done.**

*p. 9, l. 18: What do you mean with "any specific year"? Does that mean that the boundary conditions are disregarded? Can 1900 serve as a prior for 1999?*

**We deleted "any specific year". We used 1900 for 1900. It was meant that we could calculate several years at the same time not in a sequence. making the algorithm even faster.**

*p. 10, l. 13: "a more consistent"?*

**Changed to "realistic"**

*p. 10, l. 30: There is another important difference to traditional CFR techniques (by the way: spell out), namely that data assimilation at least formally does not require calibration and thus is less sensitive to stationarity issues.*

**Done. We added the comment.**

*p. 11, l. 1: "full atmosphere-ocean interaction".*
**Done.**

*p. 11, l. 25: Not only model errors, also the observation error is an issue.*
**Given that we know the "true" state, the observation error is known in our OSSE. But in real word application this is true and is discussed in outlook.**

**3  Answer to Editor**

*Dear Authors, Thanks for posting your responses to the reviewers' comments. The reviewers raised substantial points but the suggestions included in your answers to take them into account appear reasonable to me at this stage (I have not checked the new version of the paper attached to one of your comments as the revised manuscript should be submitted separately). I would thus be happy to consider a revised version for publication in Climate of the Past. My only significant concern from your answers is the issue of the soil moisture. In particular, I do not understand your answer 'We consider that this approach would reduce the consistency of the simulations, given that the moisture values considered by SPEEDY parametrizations would be still the climatological ones and not the ones produced by the CPC Leaky Bucket'. Does it mean that you are using a climatological soil moisture in Speedy? Would this imply that in SPEEDY the soil moisture is not consistent with interannual variations in precipitation? This would then be very instructive to test what would be the impact on your results of using the precipitation from SPEEDY to compute the soil moisture appplied in VS LITE. Sincerely, Hugues Goosse*

**According to your and reviewer one's suggestions we implemented the Leaky Bucket Model ([Huang et al., 1996b]) in our DA code. The Leaky Bucket Model code was extracted from VS-lite v2_3 (`ftp://ftp.ncdc.noaa.gov/pub/data/paleo/softlib/vs-lite/`). Instead of using climatological soil moisture for VS-Lite, the precipitation and temperature output from SPEEDY is used by Leaky Bucket Model to produce the new set of soil moisture with interannual variations. In the next step we repeated the off-line data assimilation runs for two VS-Lite presentations (VSL-Prod and VSL-Min). Accordingly we will add the Figures 12, 13 and 14 (in this answer Fig.1-3) to the new version of manuscript. The results show that using the new set**

a)

[Figure]

Figure 1: Global ensemble mean for analysis constrained by pseudo-TRW observations for a) VSL-Min with the climatological soil moisture (red), with the soil moisture computed by Leaky Bucket Model (blue) and free run (black); b) VSL-Prod with the climatological soil moisture (green), with the soil moisture computed by Leaky Bucket Model (blue) and free run (black). Horizontal lines exhibit the mean values. Right panels exhibit the histograms of the time-series.

of soil moisture has improved the error reduction of VSL-Min with minor improvement for VSL-Prod in both time evolution and maps of RMSE. Thus, the RMSE of VSL-Min reaches the one of VSL-Prod in the new runs. This is more clear in the RMSE maps (Fig. 2 of this answer). We added subsecton 3.2.4 to the new version of the manuscript.

Figure 3 shows the histograms of the RMSE time-series. The results show that the VSL-Min is more sensitive to the choice of soil moisture and using the calculated soil moisture with the Leaky Bucket Model improves the performance of the model. However, for VSL-Prod the improvement in error reduction 
[revised manuscript text omitted]

**3.1.2**

5

10 ~~appears as a very convenient quantity to assess the real benefit of performing DA. Figure ?? shows the existence of considerable error reduction for temperature and moisture in some geographical areas, whereas for u-wind $\varepsilon^{\text{FREE}}_{\text{REDUCTION}}$ exhibits negligible values as it is expected from the analysis of figure ??. This absence of observational constraint on the wind variables was common to all of the simulations and accordingly wind-related quantities will not be analyzed hereafter.~~

15 ~~that the error reduction regarding the free ensemble run appears modulated by the magnitude of the yearly internal variability of the particular variable at a specific site (compare figures ?? and ??). As a consequence, stations located in areas of strong yearly internal variability are more efficient than the others at reducing the error of the TA state estimate. An example of this are the stations located in Alaska which constrain temperature considerably more strongly than the ones laying on south-east USA, south America or south Africa. This finding can then be utilized as guidance for the design of optimal TRW chronology~~
20

~~An additional relevant feature of figure ?? is that, both for temperature and humidity, the error reduction is strongly localized around data-rich areas. This behavior can be explained by the negligible error reduction obtained for all the forecast variables (not shown). 
[revised manuscript text omitted]

5 ~~The use of as observation operator appears compatible with the -based technique utilized, as it is evidenced by the low RMSE levels observed in figure **??** around the observational network. Nonetheless, due to the strong nonlinear features of VSL-Min, the performance of filter is expected to be degraded with respect to the ensemble runs constrained with TA linear observations (see AC15) . This behavior can be readily seen in figure **??**, which show considerable error increases due to observation nonlinearities. Regarding temperature, $\varepsilon_{\text{INCREASE}}^{\text{NonLinObs}}$ presents particularly high values over the Labrador peninsula, central Europe~~

10

~~An interesting feature of figure **??** is the presence of unobserved zones with negative $\varepsilon_{\text{INCREASE}}^{\text{NonLinObs}}$ values for temperature, e. g. , Antarctica and for humidity over the Arabian peninsula; Indian subcontinent and southeastern China, which implies that the estimation of the TA state is not optimal over these regions. This phenomenon might be attributed to significant non-Gaussianity~~

15

**3.2.3**

[revised manuscript text omitted]

---

## Referee Report (RR1)

Response to Authors (2)

The authors have done a fine job addressing my comments and I feel the manuscript may be suitable for publication after addressing several large remaining issues with the text. I'm suggesting further revisions for additional problems with language and brevity.

I would like to thank the authors for a much clearer and better-written manuscript and for addressing our concern about the time-varying soil moisture through further work, and for making a large effort to revise and add to the text for clarity and flow. However, the manuscript is still lacking transitions and sufficient detail, and is still too brief and in many places very confusing to follow. It is still a bit jarring to read and I think the abstract in particular could be more motivating and clear regarding the new science this work has added. The sectioning is over-done, and there are multiple places where you have a single sentence constituting an entire paragraph without any transition between them, an issue I highlighted in my first revision.

For the new sections added, I couldn't take the time once again to heavily edit for language; please do go back over the additional sections to ensure that your sentence structure and word choices are sound. Guide the reader slowly through what you did and motivate it clearly. Some of the word choices and arguments are still awkward and hard to understand. A few that I caught are listed below. Please make appropriate edits throughout. Especially in the results and Discussion, I was really lost.

rephrase "Our knowledge of the climate system…governing the evolution of the oceans and atmosphere."
"state of the flow" is too vague. Be specific!
delete comma after (forecast),
delete "Furthermore," and start sentence with The
'adjoint model' – unclear on what you mean here

Could you add some more transitional sentences to guide the reader through the subsections in Section 2?

Be careful about extra-short paragraphs that only have one sentence…. Combine where appropriate.

as 'a' consequence, not 'the' , add 'any one observation may present ….with distant ones'

2.1.1 line 23: what is the 'sensor' – I don't think you have defined this yet…

Once again, the line numbering in this text just changes from 5-30 throughout which made it very difficult to give line-by-line comments.

23-25 the wording of this sentence is a bit confusing.  You're trying to say that rain gauges and anemometers take hourly-scale measurements but paleoclimate data contain a time-averaged signal. It reads as if you're saying they're all the same. Revise for clarity.

Page 6:
revise "comprising of a dynamical model" …. 'all which interact with each other'

2.2.1 You need to define the V-S Lite acronym on first use and spell it out in this title. You do that later in the text at the moment and it's out of place. It needs to be here.

what is PLF ? Redefine, the reader has forgotten.

8: grown = growth
15: change definition of FL acronym to main text, not just in the title.
17: delete 'applied'
18: change to "FL has been applied in ecological …
22: correspond = corresponds (add s)

Page 8, make line 8 into two sentences. Equation 15. Then, ….

2.4 Page 9
Change to "Experimental Design"
You define VSL here but it should be on the previous page.
change model to 'modeling'

Page 10:
10: boundaries THE model requires…
why is there a bullet here?

Page 11:
'lowest level of the state vector' – this is too much jargon. Do you just mean surface temperature?
12: what is meant by 'shifting of recorded variable?' unclear—revise

Delete Section 2.4.2 and move all of that paragraph to 3 Results.
19: change wording "We focus our analysis on temperature due to the larger error reduction in this field as compared to other variables…."

23: delete parentheses around near surface temperature and add "of near-surface temperatures"

Page 12 typo line 26 "acenso"
Page 13 at the very top here you introduce a PRESCRIBED experiment for the first time. There must be some problem with your LateX file here—You meant to have two bullets above, one with SLAB and one with PRESCRIBED, but one got lost….correct above.
Do not put a new paragraph between lines 10 and 11…. You're still discussing the same figure.

You've got LaTeX error instances of "DIFdelbegin" and end throughout the text…

comma splice. This is also a good example of a long and convoluted sentence that appears throughout that is hard to follow and understand.

20-25 this is a bit of a weak statement regarding optimal network design. We already know that trees at higher latitudes are going to be more sensitive to changes in temperature, so saying that we should measure trees in Alaska instead of South Africa isn't very helpful. We're actually desperate for more terrestrial records from the Southern Hemisphere….I know what you're trying to say here, but you need to make a better argument.

contains comma splices change to "This study, where a DA method…."

I think you mean to say "constraint on the forecast motivates us to perform our DA experiments in an offline regime"

delete comma after Prod

25-30 you say that a previous study supports Online …. But then you say "therefore we apply offline" – this doesn't follow or make sense. Review previous work and then compare it to what you did and say how it's different.  Also, "performs a more realistic temporal varialbity" makes no sense—rephrase: 'simulates more realistic ..'    and also, temporal variability of what?

Line 3: what system are you referring to?

**Ever since the results section started I have felt lost trying to follow the writing. I also feel the sections are hyper-sectioned….out to four subsections is unnecessary. Just have two paragraphs and introduce them for 3.2.2.1, 3.2.2.2.**

what is an elbow? Use mathematical language. Inflection point? What is Free run and why is it capitalized?

Your sections just jumped from 3.2.3 to 3.2.1.

Re-label as 'Time-Varying Soil Moisture"

I don't know what the 'new set' is—you have to be specific. New set is the "time-varying soil moisture fields." Spell it out.

I think when you say 'time evolution….' Do you mean the time series of global mean temperature? As in, the skill over time vs. the spatial skill? You need to clearly differentiate between these two metrics throughout the text. They are calculated differently and have different implications.

"improvement in the error reduction" – don't you just mean "error reduction" ?

rephrase "this methodology can be applied to techniques in Optimal Sensor Placement…"

effectiveness, not effectivity (that is not a word)

Line 2, "has yet to be investigated"

5: you're talking about structural biases and how they carry forward using PSMs with GCMs in a DA framework. This is discussed at length in Dee et al., 2016, and you might reference that here.

10: 'indiscriminant'

17-20 single, long, confusing sentence, single paragraph.

21-35 this section reads as a regurgitation of previous work that is not cohesively linked to what you did and with the writing. I cannot follow this text at all. "enter an offline regime" – don't you just mean "use an offline regime?" I don't understand the argument that you can't get observational constraints for internal variability with annual resolution records. Please clarify this argument dramatically.

4.2 on page 21… you do not effectively review your new results or your new work here at all…. What is meant by pollutions? "in the face of" is too colloquial. In this discussion, and the outlook, you need to provide us with a concise summary of what you did, your new findings and how it compares to previous work, outline caveats, and then give suggestions for future work and the importance of yours. At present, it is too disjoint and I think the over-sectioning is a culprit. We don't need multiple sections here. Just "Discussion."

---

## Author Response (AR2)

**Final answer to the Comments**

Acevedo et al.,

March 24, 2017

Dear Prof. Goosse and Dear reviewer #2,

Thank you so much for your constructive comments. In a new version of the manuscript we tried to remove unnecessary technical details and describe our motivations and findings in a clear format. We provided a track changes version of the manuscript at the end of this answer (Red is deleted and blue is added). We reply to all your comments here:

**1 Answer to Reviewer 2**

We wish to thank you so much for your constructive review. It would be our pleasure to do all the modifications and make the improvements you have suggested, in the next version of the manuscript. We answer your comments (*italic*) point by point (**Bold**):

*The authors have done a fine job addressing my comments and I feel the manuscript may be suitable for publication after addressing several large remaining issues with the text. I'm suggesting further revisions for additional problems with language and brevity. I would like to thank the authors for a much clearer and better-written manuscript and for addressing our concern about the time-varying soil moisture through further work, and for making a large effort to revise and add to the text for clarity and flow. However, the manuscript is still lacking transitions and sufficient detail, and is still too brief and in many places very confusing to follow. It is still a bit jarring to read and I think the abstract in particular could be more motivating and clear regarding the new science this work has added. The sectioning is over-done, and there are multiple places where you have a single sentence constituting an entire paragraph without any transition between them, an issue I highlighted in my first revision. For the new sections added, I couldn't take the time once again to heavily edit for language; please do go back over the additional sections to ensure that your sentence structure and word choices are sound. Guide the reader slowly through what you did and motivate it clearly. Some of the word choices and arguments are still awkward and hard to understand. A few that I caught are listed below. Please make appropriate edits throughout. Especially in the results and Discussion, I was really lost.*

**We agree. In the current version, the Discussion, abstract and introduction have been heavily edited. Now we open our main scientific questions in introduction and close them in Discussion. Regarding the introduction we give more citations on the previous studies around the forward modeling and its different applications as well as similar DA studies recently published. Regarding the discussion we merged the subsections together and formed a homogeneous story of our main results.**

*Page 1*

*8 rephrase "Our knowledge of the climate system...governing the evolution of the oceans and atmosphere."*

*9 "state of the flow" is too vague. Be specific!*

*12 delete comma after (forecast),*

*12 delete "Furthermore," and start sentence with The*

*21 'adjoint model' – unclear on what you mean here Could you add some more transitional sentences to guide the reader through the subsections in Section 2? Be careful about extra-short paragraphs that only have one sentence.... Combine where appropriate.*

**We merged many subsections and edited the whole section 2. We start with the Data Assimilation basics, KF, EnKF and time-averaged EnKF. Then we introduced the forward model representations along with the concept of Fuzzy Logic. Finally we presented our simulation design.**

*Page 5*

*12 as 'a' consequence, not 'the' , add 'any one observation may present ....with distant ones'*

*2.1.1 line 23: what is the 'sensor' – I don't think you have defined this yet... Once again, the line numbering in this text just changes from 5-30 throughout which made it very difficult to give line-by-line comments.*

*23-25 the wording of this sentence is a bit confusing. You're trying to say that rain gauges and anemometers take hourly-scale measurements but paleoclimate data contain a time-averaged signal. It reads as if you're saying they're all the same. Revise for clarity.*

**Given that the "instantaneous" and "time-averaged" variables are frequently used by the climate community, we removed the sentence describing these two terms in the new version.**

*Page 6:*

*3 revise "comprising of a dynamical model" .... 'all which interact with each other'2.2.1 You need to define the V-S Lite acronym on first use and spell it out in this title. You do that later in the text at the moment and it's out of place. It needs to be here. 24 what is PLF ? Redefine, the reader has forgotten.*

**We have moved the text describing VSL into the introduction where we introduce TRW forward model as well as PLF.**

*Page 7*

*8: grown = growth*

*15: change definition of FL acronym to main text, not just in the title.*

*17: delete 'applied'*

*18: change to "FL has been applied in ecological ...*

*22: correspond = corresponds (add s)*

**Done.**

*Page 8, make line 8 into two sentences. Equation 15. Then, ....*
*2.4 Page 9*
*Change to "Experimental Design"*
*You define VSL here but it should be on the previous page.*
*16 change model to 'modeling'*

**Done.**

*Page 10: 10: boundaries THE model requires...*
*18 why is there a bullet here?*
*Page 11:*
*5 'lowest level of the state vector' – this is too much jargon. Do you just mean surface temperature?*
*12: what is meant by 'shifting of recorded variable?' unclear—revise*
*Delete Section 2.4.2 and move all of that paragraph to 3 Results.*
*19: change wording "We focus our analysis on temperature due to the larger error reduction in this field as compared to other variables...."*
*23: delete parentheses around near surface temperature and add "of near-surface temperatures" Page 12 typo line 26 "acenso"*
*Page 13 at the very top here you introduce a PRESCRIBED experiment for the first time. There must be some problem with your LateX file here—You meant to have two bullets above, one with SLAB and one with PRESCRIBED, but one got lost....correct above.*

**Yes there were some problems with the latexdiff. Now it is corrected. The bullets are for the two ocean configurations.**

*Page 14*
*Do not put a new paragraph between lines 10 and 11.... You're still discussing the same figure. You've got LaTeX error instances of "DIFdelbegin" and end throughout the text...*
*17 comma splice. This is also a good example of a long and convoluted sentence that appears throughout that is hard to follow and understand.*
*20-25 this is a bit of a weak statement regarding optimal network design. We already know that trees at higher latitudes are going to be more sensitive to changes in temperature, so saying that we should measure trees in Alaska instead of South Africa isn't very helpful. We're actually desperate for more terrestrial records from the Southern Hemisphere....I know what you're trying to say here, but you need to make a better argument.*
*23 contains comma splices*

**We edited the text. We added your comment as an extra sentence about the terrestrial records from the Southern Hemisphere.**

*25 change to "This study, where a DA method...." 32 I think you mean to say "constraint on the forecast motivates us to perform our DA experiments in an offline regime"*

**These sentences are already appearing in the discussion with a new format.**

*Page 15*
*10 delete comma after Prod*
*25-30 you say that a previous study supports Online …. But then you say "therefore we apply offline" – this doesn't follow or make sense. Review previous work and then compare it to what you did and say how it's different. Also, "performs a more realistic temporal varialbity" makes no sense—rephrase: 'simulates more realistic ..' and also, temporal variability of what?*

**Given that our DA attempt appeared to be in an "off-line regime", in the next step we have conducted an "off-line DA". There is a difference between "off-line regime" and "off-line DA method" which we made clearer in the new manuscript. The first one indicates that the forecast state of the DA has no skill (a DA with reinitialization after analysis) but the second is done using the forecast from free ensemble in an off-line strategy with no reinitialization. We have changed the sentence to "They concluded that in the off-line method temporal consistency of the model is lost."**

*Ever since the results section started I have felt lost trying to follow the writing. I also feel the sections are hyper-sectioned….out to four subsections is unnecessary. Just have two paragraphs and introduce them for 3.2.2.1, 3.2.2.2. Page 17?? 3 what is an elbow? Use mathematical language. Inflection point? What is Free run and why is it capitalized?*
*Your sections just jumped from 3.2.3 to 3.2.1. Re-label as 'Time-Varying Soil Moisture" 12 I don't know what the 'new set' is—you have to be specific. New set is the "time-varying soil moisture fields." Spell it out.*
*13 I think when you say 'time evolution….' Do you mean the time series of global mean temperature? As in, the skill over time vs. the spatial skill? You need to clearly differentiate between these two metrics throughout the text. They are calculated differently and have different implications.*
*19 "improvement in the error reduction" – don't you just mean "error reduction" ? 28 rephrase "this methodology can be applied to techniques in Optimal Sensor Placement…" 29 effectiveness, not effectivity (that is not a word)*
*Page 18*
*Line 2, "has yet to be investigated"*
*5: you're talking about structural biases and how they carry forward using PSMs with GCMs in a DA framework. This is discussed at length in Dee et al., 2016, and you might reference that here. 10: 'indiscriminant'17-20 single, long, confusing sentence, single paragraph.*
**As described, the subsubsections are removed and the Result section is modified.**
*21-35 this section reads as a regurgitation of previous work that is not cohesively linked to what you did and with the writing. I cannot follow this text at all. "enter an offline regime" – don't you just mean "use an offline regime?" I don't understand the argument that you can't get observational constraints for internal variability with annual resolution records. Please clarify this argument*

*dramatically.*

We agree that the previous version did not describe clearly the concept of "off-line regime" and "off-line DA". Now in the Results and Discussion section of the new version we described these two very different concepts more clearly. If the forecast state of the DA shows no skill but the analysis, then we are in the "off-line regime", although we are doing an "on-line DA". An "off-line DA" means that we do not reinitialize the model when the observation is available. We use the free ensemble forecast (the ensemble run without assimilation) for that time to produce the analysis state. Generally, we do "on-line DA" to have skill in forecast which is dynamically consistence not the analysis. However, we showed that in our experiment using SPEEDY we have no skill in forecast. We hope that the new version of the manuscript can clarify these two concepts.

*4.2 on page 21... you do not effectively review your new results or your new work here at all.... What is meant by pollutions? "in the face of" is too colloquial. In this discussion, and the outlook, you need to provide us with a concise summary of what you did, your new findings and how it compares to previous work, outline caveats, and then give suggestions for future work and the importance of yours. At present, it is too disjoint and I think the over-sectioning is a culprit. We don't need multiple sections here. Just "Discussion."*

We agree and we have modified the manuscript according to your comment.

**2 Answer to Editor**

*Dear Authors, Thanks for submitting the revised version of your manuscript. Both reviewers agreed that this revised version is significantly improved compared to the initial one but one reviewer considers that major changes are required, in particular because the text is still hard to follow. I agree with this evaluation and additional revisions following reviewers' suggestions are thus required, in particular in order to improve the clarity of the text, before a potential publication in Climate of the Past. Best regards, Hugues Goosse*

Dear Prof. Goosse,
thank you very much for consideration of the manuscript. We have edited the text according to the suggestions of the reviewer and we hope that the new version of our manuscript is easier to follow and ready for publication in Climate of the Past.

Best regards,
on behalf of all the co-authors,
Bijan Fallah

[revised manuscript text omitted]

---

## Author Response (AR3)

**Answer to suggestions from the Editor**

Acevedo et al.,

April 25, 2017

Dear Prof. Goosse,

Thank very much for your suggestions. We took them all into consideration for the last version of our manuscript, as you can see in the following suggestion/reply list. Additionally, we made a final effort to improve the readability of the manuscript. We removed unimportant technical details, such as the comparison between prescribed and slab ocean configurations which is not relevant for the focus of the paper, and with that the last 2 sections of the paper got highly improved.

*Abstract, Line 9-10: It is not clear to me to which result you refer to in the sentence: 'the error reduction achieved by assimilating a pseudo-TRW chronologies is modulated by the strength of the yearly internal variability of the model'. Additionally, the word 'strength' does not seem appropriate in the sentence.*
**Here we refer to all our experiments, accordingly we use now "in general" instead of "Additionally".**

*Additionally, the word 'strength' does not seem appropriate in the sentence.*
**'strength' changed by 'magnitude'.**

*Page 1, Line 17. Please specify why 'paleoclimate modeling ideas have been proposed'. The goal was to solve which problem?*
**Taken. Middle part of the paragraph refrased:** *To the present, many different ideas have been proposed in order to link proxy records to the paleoclimate conditions where they were created, e.g., data-driven statistical techniques, climate model hindcasts and Bayesian probabilistic methods (see Crucifix (2012) as a recent review). Among this plethora of approaches, DA methodologies are today particularly appealing as they deliver estimates of paleoclimate quantities, by systematically combining the information of paleoclimate records with the dynamical consistence of climate simulations (Brönnimann, 2011; Hakim et al., 2016).*

*Page 2, line 2 'as reference'. Did you mean 'for additional references' or 'for a review'?*
**we meant 'for reviews'. Changed accordingly**

*Page 2, line 9. Please specify at this stage what you mean by 'beginning the off-line condition'.*

sentence rephrased: *Furthermore, some recent studies have assumed the presence of the off-line condition, and accordingly have removed the reinitialization step after assimilation (Steiger et al., 2014; Dee et al., 2016; Hakim et al., 2016)*

*Page 2, line 24. 'an online EnKF scheme'. If I am right, 'on line' has not been defined yet.*

Now it is explicitly defined: *These type of DA methodologies will be referred to in this paper as "off-line DA techniques", in order to contrast them with traditional "on-line DA techniques", where the state of the model is updated after the assimilation of observations.*

*Page 2, line 21. What do you mean by 'in the realism of DA-based climate reconstructions' ? Please rephrase*

That is an error coming from the editing process, Removed

*Page 2, line 28. 'the both', suppress 'the'*

Suppressed.

*Page 4, line 24. I would not use 'aspect' in this context.*

Agreed, the beginning of the paragraph was rephrased.

*Page 6, Eq. 14. tn and tau are not defined if I am right.*

Now they are defined.

*Page 10, line 22. ENSO is not an annular mode.*

That was a mistake, now removed.

*Page 11, line 14. 'analysis is more evident in the RMSE maps'. I do not see what you refer to here.*

That was a mistake, now removed.

*Page 11, lines 24-26. 'However, the use of VSL-Prod instead of VSL-Min appears beneficial to the filter performance for the analysis, as demonstrated in Figure 6b. The expected value of the RMSE shifts significantly toward lower values for VSL-Prod compared to the free ensemble run.' I do not follow why you first compare VSL-Prod and VSL-Min in one sentence and then VSL-Prod and Free in the next one.*

That was very confusing. This subsection was simplified and rephrased. Now we believe it is much more understandable

*Page 12, Fig.9. Is figure 9 for the 'non-cycling' method? It is not clear from the caption.*

All the captions now mention explicitly whether on-line or off-line DA was used.

Finally, we would like to thank you again for the thorough editing process you followed with our paper, which considerably increased the clarity of the manuscript.

Best regards,

on behalf of all the co-authors,
Bijan Fallah